# Impact of global climate cooling on Ordovician marine biodiversity

Daniel Eliahou Ontiveros [1] ✉, Gregory Beaugrand [1], Bertrand Lefebvre [2], Chloe Markussen Marcilly[3], Thomas Servais[4] & Alexandre Pohl [5] ✉

Global cooling has been proposed as a driver of the Great Ordovician Biodiversification Event, the largest radiation of Phanerozoic marine animal Life. Yet, mechanistic understanding of the underlying pathways is lacking and other possible causes are debated. Here we couple a global climate model with a macroecological model to reconstruct global biodiversity patterns during the Ordovician. In our simulations, an inverted latitudinal biodiversity gradient characterizes the late Cambrian and Early Ordovician when climate was much warmer than today. During the Mid-Late Ordovician, climate cooling simultaneously permits the development of a modern latitudinal biodiversity gradient and an increase in global biodiversity. This increase is a consequence of the ecophysiological limitations to marine Life and is robust to uncertainties in both proxy-derived temperature reconstructions and organism physiology. First-order model-data agreement suggests that the most conspicuous rise in biodiversity over Earth's history – the Great Ordovician Biodiversification Event – was primarily driven by global cooling.

Following the appearance of most animal phyla during the "Cambrian Explosion"[1], the Ordovician reflects their rapid diversification at lower taxonomic levels during the "Great Ordovician Biodiversification Event" (GOBE)[2]. This radiation constitutes the most prominent increase in the biodiversity of marine organisms during the entire Phanerozoic[3,4]. It was already identified in the seminal work of Sepkoski[5] and stands in most recent compilations[3,4], although the timing of its onset is strongly dependent on the studied taxonomic group and geographical region considered[6].

The causes of the GOBE remain largely unconstrained and several hypotheses are debated. A first hypothesis invokes a decrease in tropical ocean temperatures from levels deleterious to marine organisms[7] (>40 °C) in the Cambrian to modern-like values (ca. 30 °C) in the Early and Middle Ordovician[8]. This hypothesis is supported by updated temperature proxy compilations[9,10]. A second hypothesis proposes that the fragmentation of supercontinent Rodinia and related eustatic sea-level rise may have created new ecological niches[2,11]. A third hypothesis, based on the analysis of $O_2$-dependent fractionation of carbon isotopes during photosynthesis, suggests a long-term oxygenation of the early Paleozoic atmosphere[12]. A fourth hypothesis suggests that frequent impacts on Earth of kilometer-sized asteroids may have created new ecological niches[13]. A last hypothesis proposes that the observed increase in phytoplankton biodiversity may have modified trophic structures and fostered the diversification of plankton-feeding groups[14].

Recently, the extraterrestrial hypothesis has been rejected based on updated dating of geological strata[15]. In addition, evidence has accumulated that atmospheric oxygen concentrations may have already been relatively elevated during the Cambrian and Early Ordovician[16–18], and proxies for the global extent of anoxia testify of stable ocean redox during the main phase of the GOBE[19], together ruling out oxygenation as the main driver of biodiversification. Question remains, whether the GOBE may result from extrinsic (i.e., paleogeographical and/or climatic) or intrinsic (i.e., biological) drivers.

[1]Univ. Littoral Côte d'Opale, CNRS, Univ. Lille, UMR 8187 LOG, F-62930 Wimereux, France. [2]Univ Lyon, Univ Lyon 1, ENSL, CNRS, LGL-TPE, F-69622 Villeurbanne, France. [3]Centre for Earth Evolution and Dynamics, University of Oslo, 0315 Oslo, Norway. [4]Univ. Lille, CNRS, UMR 8198-Evo-Eco-Paleo, F-59000 Lille, France. [5]Biogéosciences, UMR 6282 CNRS, Université de Bourgogne, 6 Boulevard Gabriel, 21000 Dijon, France. ✉e-mail: danyeo@hotmail.fr; alexandre.pohl@u-bourgogne.fr

Here we use a coupled paleoclimatic and macroecological model[20,21] to evaluate the individual contributions of extrinsic mechanisms, namely the long-term Ordovician cooling trend[8–10] and paleogeographical evolution[11,22], to the trajectory of marine biodiversity during the Ordovician (between 490 to 430 Ma). In contrast with previous attempts based on temporal correlations, our approach allows us to quantify processes and establish causal relationships.

## Results and discussion

### Global cooling triggers Ordovician biodiversification

Ocean temperatures were simulated using the Fast Ocean Atmosphere Model (FOAM), a mixed-resolution ocean-atmosphere general circulation model[23] (Methods). We conducted one simulation every 10 Myrs from 490 Ma to 430 Ma. Solar luminosity follows the model of stellar physics of ref. [24] and, in our main experiments, we varied atmospheric $pCO_2$ to reproduce the long-term Ordovician cooling trend of the oxygen isotope compilation of ref. [10]. As a result, tropical sea-surface temperatures decrease from ca. 45 °C at 490 Ma to ca. 30 °C at 430 Ma. Importantly, FOAM has been shown to satisfactorily capture the latitudinal seawater temperature gradient during both Paleozoic icehouse periods (Ordovician glaciation[25]) and Mesozoic thermal maxima (Oceanic Anoxic Event 2, see Peer Review file published with ref. [26]).

Biodiversity was simulated using a macroecological model based on the interaction between many modeled pseudo-species and their environment, which captures biodiversity patterns well in the modern ocean[21] (Methods). Here we forced it with the ocean temperatures simulated with FOAM (Methods). A similar coupling has previously been shown to successfully capture the spatial-temporal biodiversity patterns of early Paleozoic acritarchs[20]. In the model, thousands of pseudo-species were generated, each one occupying a unique thermal niche *sensu* Hutchinson[27] based on the principle of exclusive competition[28]. Although the number of ecological niches can be decoupled from the number of species under certain circumstances in reality[29], we note that this and similar modeling approaches have previously been proven to satisfactorily simulate biodiversity at the global scale amongst groups of organisms of various complexity, both on land and in the ocean[30,31]. In each model grid point, biodiversity is defined as the number of pseudo-species whose thermal niche overlaps the mean annual ocean temperature. The lowermost ($t_{min} = -1.8$ °C) and uppermost ($t_{max} = 44$ °C) temperature bounds of the niches were chosen to provide the best simulation of biodiversity spatial patterns in the modern ocean[21]. We further note that the lower bound is physically imposed as the temperature at which seawater freezes, while the upper bound is close to the value of 41 °C derived from the analysis of the temperature limits of tropical marine ectotherm species of ref. [7]. We focused our analysis on the continental shelves and excluded both the deep ocean (here defined as all grid points deeper than −250 m) and polar (>60 ° latitude) regions since they are not well represented in paleontological databases[20,32,33].

Analysis of the spatial patterns reveals that the maximum of simulated biodiversity (or equivalently, maximum number of niches available) shifts from high latitudes (60° S) between 490 Ma and 470 Ma, to middle latitudes at 460 Ma, and finally to low latitudes (from 450 Ma onward) (Fig. 1). This spatial-temporal pattern arises from the interaction between the modeled pseudo-species and their environment, with lethally high ocean temperatures making the low-latitude shelves hostile to marine life in the late Cambrian and gradually more suitable as global climate cools. This result, i.e., that global cooling triggers biodiversification in the model, arises in part from the niche-temperature interaction and a thermal mid-domain effect[34,35]; ecological niches being defined between minimum and maximum temperature bounds, a maximum number of niches overlaps intermediate temperatures, resulting in maximum biodiversity around 23 °C (ref. [35]). In the absence of better constraints on the ecophysiology of marine organisms in the deep past, our approach necessarily

relies on the assumption that this biodiversity–temperature relationship did not change over geological time. While such assumption may seem limiting, we note that the biodiversity–temperature relationship emerging from our model is supported by the mechanistic model of ref. [36] that demonstrates that the diversity-temperature relationship is expected, on grounds of thermodynamics and physiology principles, to peak at moderate temperatures in the range of 15–25 °C and decrease outside of this range.

The increase in marine biodiversity simulated during the Ordovician stands when the cooling scenario is varied in the model, although the specific shape of the temporal trend in biodiversity depends on the details of the temperature reconstruction[9,37] (Supplementary Figs. 1 and 2). The possible underestimation of the magnitude of Ordovician cooling arising from the heterogeneous spatial distribution of temperature proxies[38] suggests that the simulated increase in biodiversity may be underestimated as well. Our model does not capture the Late Ordovician Mass Extinction (ca. 445 to 443 Ma), which was caused by a combination of factors that are not accounted for here, including Hirnantian sudden cooling, sea-level fall and possibly anoxia[25,39,40].

Model results were compared with three different fossil global biodiversity curves, mainly composed of brachiopods, trilobites and conodonts (Fig. 2). Differences between these curves arise from the variable temporal resolution, with the seminal curve of Sepkoski et al. (ref. [5]) exhibiting the lowest resolution (>5 Myrs). Differences also reflect the use of different datasets. Rasmussen et al. (ref. [41]) based their compilation on the Paleobiology Database (PBDB), which mostly includes data from Laurentia and Baltica (>50% in 2019), while the Geobiodiversity Database[3] (GBDB) mainly includes Chinese data. In our simulations, species richness monotonically increases in response to global climate cooling from the late Cambrian (490 Ma) to the Katian (Late Ordovician, 450 Ma) and subsequently stabilizes between 450 Ma and the early Silurian (430 Ma), in first-order agreement with the reference biodiversity curves (Fig. 2). Biological evolution may have played a role during the GOBE as well[14,42], but it is not necessary to invoke this process to simulate a first-order rise in marine biodiversity during the Ordovician in our model.

In order to facilitate the attribution of causes and quantify the respective contributions of global climate change and paleogeographical evolution to model biodiversification, we conducted additional simulations characterized by a constant tropical sea-surface temperature of ca. 30 °C (Fig. 2). These simulations constitute a sensitivity test to the continental configuration. They show that continental drift alone triggers no substantial increase in marine biodiversity and confirm that global climate cooling is the driver of the simulated marine biodiversification. The same conclusions are reached when conducting simulations under a warmer global climate with tropical sea-surface temperatures of ca. 40 °C (Supplementary Fig. 3).

### Impact of the thermal upper limit of model pseudo-species

Our main experiments are based on contemporaneous thermal limits for marine Life[21]. This possibly constitutes an actualistic bias, since Ordovician organisms may have been adapted to the warmer early Paleozoic climate. To determine to what extent model biodiversification is affected by the thermal upper limit $t_{max}$, we conducted additional simulations for alternative values of $t_{max}$. We quantified the degree of similarity between the biodiversity spatial patterns simulated under warm and cool climates, characterized by tropical sea-surface temperatures of ca. 40 °C and 30 °C. These values were chosen to represent the end-member (respectively warm and cool) climatic states of the long-term Ordovician cooling trend. The closer the results obtained under these contrasting global climatic states, the weakest the impact of global climate cooling on biodiversity. Simulations were performed using the continental configuration for 460 Ma, which

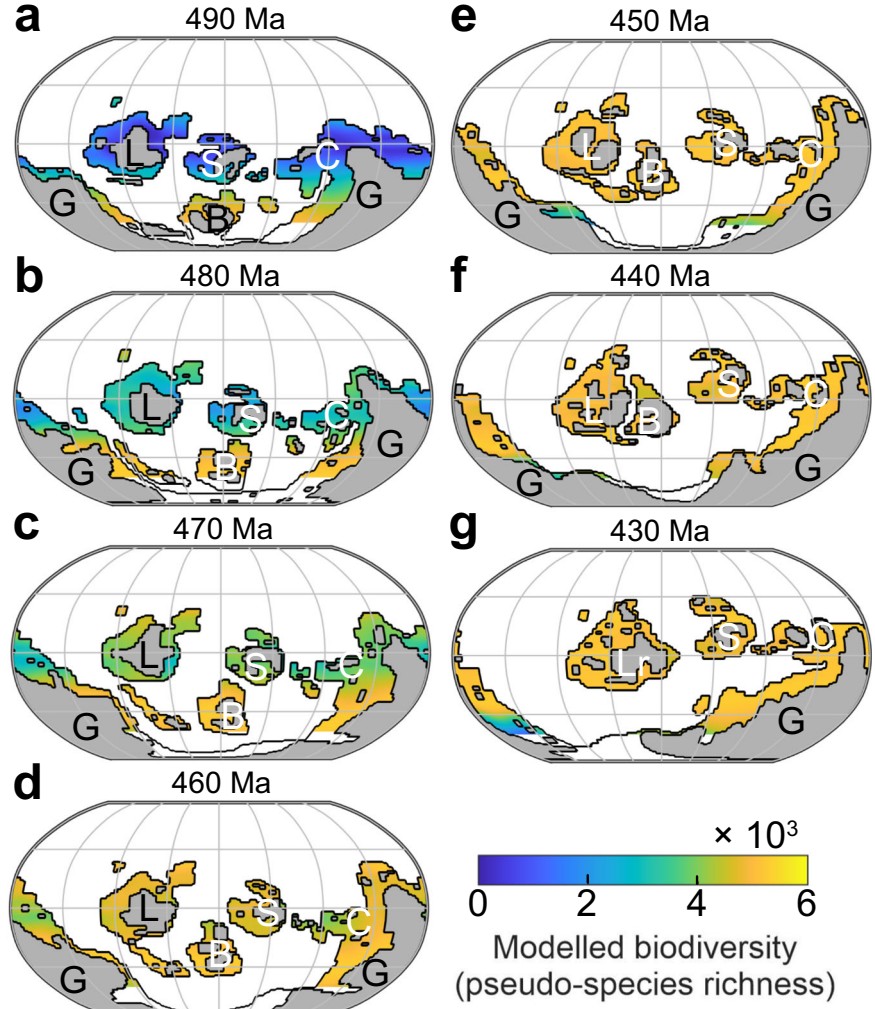

**Fig. 1 | Spatial patterns of simulated Ordovician biodiversity. a–g** Landmasses are shaded gray. Continental shelves are outlined in black. Robinson projection with parallels drawn every 30° latitude. G Gondwana, L Laurentia, B Baltica, S Siberia, C South China, Lr Laurussia (after docking of Baltica with Laurentia at 430 Ma).

represents an intermediate state between 490 Ma and 430 Ma. This sensitivity analysis is presented in a Taylor diagram (Fig. 3). Correlation is the highest for $t_{max} = 65\,°C$. Standard deviation is the closest to the reference simulation for $t_{max} = 78\,°C$. The root mean square deviation is the lowest for values of $t_{max}$ ranging between 68 °C and 78 °C. The correlation decreases very fast when $t_{max}$ decreases, and only the simulation using $t_{max} = 65\,°C$ shows a correlation exceeding 0.95. Altogether, these results demonstrate that a thermal upper limit exceeding 65 °C is necessary to suppress the impact of global climate cooling on biodiversity. This value is >20 °C above the (44 °C) modern limit used in the main experiments.

Physiological adaptation of the Paleozoic Life to such extent is unlikely. The compilation of ref. 43. shows a very limited change in the thermal tolerance of marine organisms over the last 500 million years. In addition, several lines of evidence suggest that the atmospheric oxygen concentration was, if anything[16,17], lower than today during the early Paleozoic[44,45]. Due to the increase in organism metabolism (thus dissolved oxygen consumption) with increasing temperatures, a reduced ocean oxygenation would lower the thermal upper limit for marine ectotherms compared to modern[33]. It is therefore likely that $t_{max}$ during the early Paleozoic would have been lower, rather than higher, than modern. Therefore, the increase in biodiversity simulated in response to global cooling stands as a robust model result that is poorly dependent on (reasonable) changes in organism physiology.

The temperature-biodiversity relationship also depends on the distribution of the niches in the temperature dimension. In our simulations, the niches are uniformly distributed along the temperature axis, between minimum and maximum temperatures. For a same $t_{max}$, however, an asymmetric distribution of the niches skewed towards the high temperatures would result in more species adapted to warmer climates. It could be argued that such adaptations of marine Life to warmer temperatures may have developed in early Paleozoic oceans, significantly reducing the impact of global cooling on marine biodiversity in our simulations. However, such alteration of the temperature-biodiversity relationship would increase the number, packing and ultimately overlapping of niches at high temperatures. It would conflict with the fundamental principle of competitive exclusion[28]. Therefore, we consider such changes in the temperature-biodiversity relationship as unlikely.

### Establishment of a modern latitudinal diversity gradient
The latitudinal biodiversity gradient (LBG) –the increase in biodiversity from the poles to the low latitudes– is a pervasive macroecological pattern of life on Earth[21,46]. The distribution of fossil occurrences shows that this pattern has changed over long periods of time[20,47,48]. Our simulations suggest that, following the appearance of most animal phyla during the Cambrian Explosion[1], a modern-like LBG first established during the GOBE in response to global climate cooling. During

the late Cambrian and early Ordovician (490 to 470 Ma), the maximum in modeled biodiversity is located over mid to high latitudes, contrasting with modern LBGs, while the lower latitudes are too warm to host diverse and abundant marine life (Fig. 4). In response to global climate cooling, the maximum in biodiversity gradually shifts to lower latitudes as large habitats gradually become livable for the modeled pseudo-species. The maximum in biodiversity reaches ca. 40° S at 460 Ma and covers the 40° S–20° N latitudinal band from 450 Ma onwards. The evolution of the LBG simulated during the Ordovician is robust when the model is run over the global ocean (Supplementary

Fig. 4). Therefore, results are not overly dependent on the considered continental reconstruction and position of the shallow-water shelves (see Fig. 1).

Although sampling bias may distort the LBGs reconstructed based on paleontological data and obscure model-data comparison[46], simulated spatial-temporal patterns of biodiversity are supported by the LBG of Cambrian and Ordovician acritarchs, which show a minimum in biodiversity along the equator at 490 Ma and a shift of maximum biodiversity from ~60 °S at 480 Ma to the low latitudes during the Late Ordovician[20]. Our results are also supported by the LBG independently reconstructed during the Ordovician by Kröger et al. (ref. 32) based on the Paleobiology Database[4], which shows a transition from a bimodal Early Ordovician LBG to a unimodal Late Ordovician LBG. The authors note that this evolution results from the large increase in low-latitude biodiversity during the GOBE, in agreement with the results of our coupled climate-macroecological model (Figs. 1 and 4).

Our results, obtained with a coupled climate-macroecological model featuring minimal biological assumptions, shed new light on the mechanisms that allowed the establishment of a modern-like LBG for the first time in Earth's history. They suggest that global climate change, rather than the two other usual suspects, paleogeography or history[48], was decisive. They also reaffirm the contribution of a cool global climatic state to the development of a steep LBG featuring a maximum in biodiversity at the low latitudes in line with paleontological data of the Permian-Triassic transition[47].

### Regional biodiversification patterns

Our spatially-resolved simulations allow biodiversification signals to be compared in different paleocontinents. For comparisons, we used the brachiopod database of ref. 6. (Fig. 5). Our model captures the increase in fossil brachiopod biodiversity in Laurentia, Siberia and South China (Fig. 5a–c), although with varying degrees of model-data agreement. In Laurentia and Siberia, simulated biodiversity rises too early compared to data. In South China, the general timing of biodiversity increase matches the temporal trends reconstructed based on data from the PBDB between 490 Ma and 450 Ma[6]. From 440 Ma to 430 Ma, model biodiversity stabilizes while paleontological data show a large drop in biodiversity. This discrepancy in South China probably reflects the

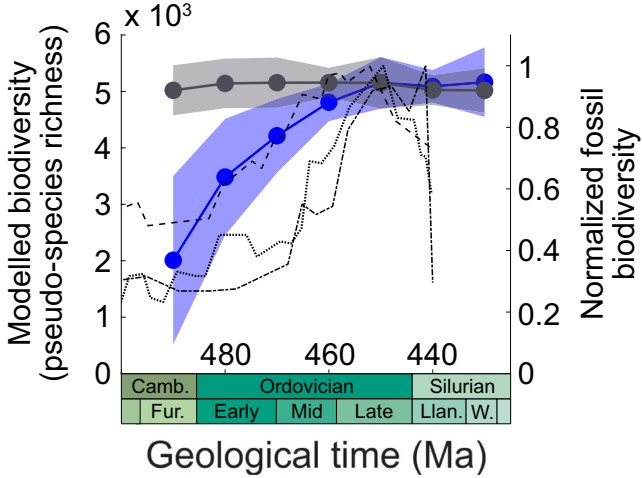

**Fig. 2 | Temporal trends in simulated and observed Ordovician biodiversity.** Model biodiversity was calculated as the median species richness simulated over continental shelves (left y-axis) ±1 standard deviation (envelope). It is shown for two scenarios: proxy-derived global climate cooling[10] (blue) and theoretical constant climatic state (sensitivity test to the continental configuration; gray). Normalized fossil biodiversity (right y-axis) is after the compilations of Sepkoski et al. (ref. 5) (dotted line), Fan et al. (ref. 3) (dashed line) and Rasmussen et al. (ref. 41) (dashed-dotted line). Geological ages are after the International Chronostratigraphic Chart v2023/04. Camb Cambrian, Fur Furongian, Llan Llandovery, W Wenlock.

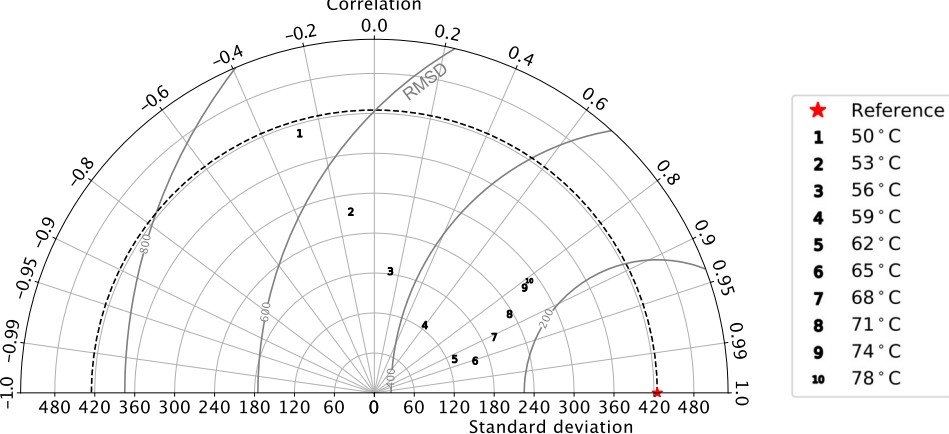

**Fig. 3 | Sensitivity analysis to thermal maximum of pseudo-species by means of a Taylor diagram.** We here compare the pseudo-species richness simulated in two dimensions (along the latitudes and longitudes) in various simulations. All simulations use the continental configuration for 460 Ma. The red star represents the standard simulation ran at 460 Ma with a tropical sea-surface temperature of ca. 30 °C (using $t_{max}$ = 44 °C as the uppermost temperature for the pseudo-species thermal niches). This simulation is used as a reference. Numbers represent individual simulations ran (at 460 Ma as well) under a warmer climate (tropical sea-surface temperature of ca. 40 °C) featuring alternative $t_{max}$ values ranging from 50 °C (simulation number 1) to 78 °C (simulation 10). The Taylor diagram, by

combining representations of the correlation coefficient and the Root-Mean-Square Deviation (RMSD) between each simulation and the reference, and the standard deviation of pseudo-species richness in each simulation, permits to investigate what values of $t_{max}$ are required to make the pre-cooling and post-cooling pseudo-species richness similar at 460 Ma, i.e., what values of $t_{max}$ are required to suppress the modeled increase in biodiversity in response to global climate cooling. It is used to demonstrate that the simulated increase in model biodiversity in response to cooling stands in the model, unless unlikely values are used for $t_{max}$ (see main text).

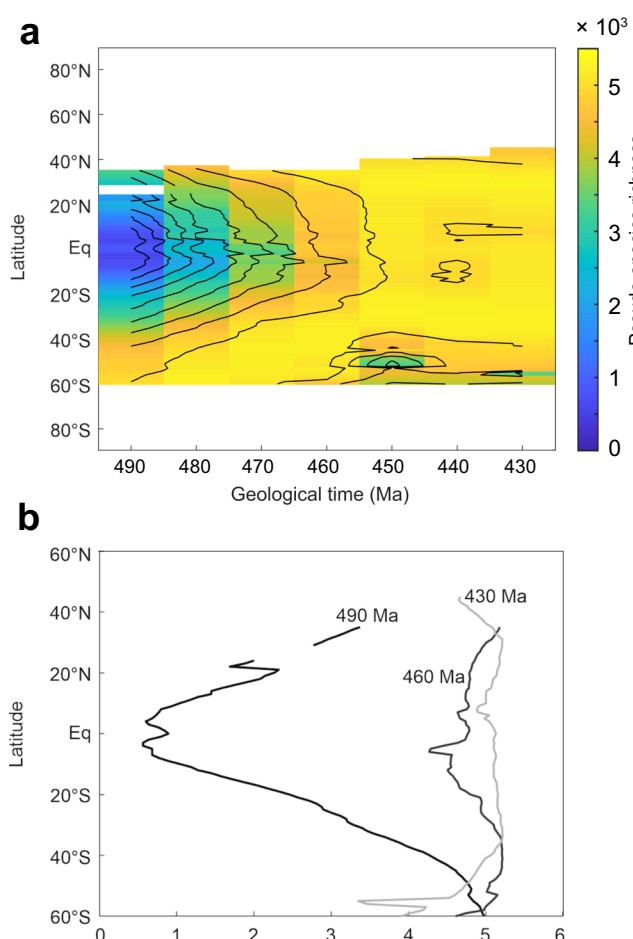

**Fig. 4 | Changes in the LBG during the Ordovician. a** Median value of modeled biodiversity per latitude vs. geological time. **b** Simulated median biodiversity for 490 Ma, 460 Ma and 430 Ma. C Cambrian, F Furongian, Llan Llandovery, W Wenlock.

absence of Late Ordovician Mass Extinction in our simulations. Model-data mismatch in Laurentia and Siberia may reflect limitations in our numerical model (including the low spatial resolution), uncertainties in temperature change during the Ordovician (see e.g. error margins of ref. 9), errors in the latitudinal position of landmasses (often exceeding 15°, see ref. 49) or biases in the paleontological database, but most probably represents a combination of all these factors.

Baltica contrasts with other continents by showing a mostly flat simulated biodiversity temporal trend, while brachiopod data exhibit a strong increase between 490 and 450 Ma (Fig. 5d). The muted model biodiversity change arises from the combination of the latitudinal drift of Baltica to the North during the Ordovician with the long-term Ordovician cooling trend, which together make Baltica closely track a climatic belt in response to global cooling (Fig. 1). As a result, temperature in the Baltica epicontinental shelves remains virtually constant, and so does simulated biodiversity. Although the same factors listed above probably contribute to model-data discrepancy, it is suspected that the continental reconstruction plays a key role here. Indeed, the uncertainty in the latitudinal position of Baltica is large in the Ordovician (Supplementary Fig. 5). Alternative paleogeographical reconstructions[50] notably exhibit a smaller latitudinal drift of Baltica –from lower latitudes at 490 Ma to higher latitudes at 430 Ma– that could help bring model and data into closer agreement (Supplementary Fig. 5). Yet, it should be noted that these reconstructions hold

even larger uncertainties in the latitudinal position of Baltica due to the lower number of paleomagnetic poles in use (3 in ref. 50. vs. 24 in the reconstructions used in our main experiments[51]). Model-data disagreement for Baltica may also reflect the inability of our model to capture finer, regional-scale environmental changes contributing to biodiversification. Indeed, quantitative analysis of brachiopod occurrences from the eastern Baltic paleobasin suggests that carbonate shelf development played a key role in regional diversification by enhancing substrate heterogeneity[52], in line with the vision that reefs are biodiversity hotspots[53].

### Extrinsic vs. intrinsic drivers of GOBE

Our results have implications for our understanding of the mechanisms that drove the GOBE. They highlight the decisive role of global climate change, although large uncertainties remain in the reconstruction of early Paleozoic ocean temperatures. Indeed, the low oxygen isotope values typifying the early Paleozoic are sometimes interpreted as reflecting very high ocean temperatures (>40 °C[8,9] at low latitudes, possibly exceeding 50 °C[10,37]) under the assumption of a constant oxygen seawater composition during the Phanerozoic –an assumption supported by coupled oxygen isotope and clumped isotope measurements that show no long-term trend in seawater $\delta^{18}O$ over the last 500 Myrs[54]. Other studies argue that the long-term trend in $\delta^{18}O$ represents variations in the oxygen isotope composition of Phanerozoic oceans[55,56]. Correction for this temporal drift results in early Paleozoic temperatures that are in the same order of magnitude as Mesozoic and early Cenozoic greenhouse climates[55] or even modern[56]. This second view is supported by the climate model-based interpretation of the distribution of the lithological indicators of climate of ref. 57. Our coupled climate-biodiversity modeling shows that a long-term cooling from very high temperature in the Cambrian to modern-like levels in the Ordovician provides a conservative, minimal scenario allowing the GOBE to take place in the model, and a modern-like LBG to emerge in agreement with paleontological evidence, even without invoking additional, biological drivers. We do not intend to mean that evolutionary innovation and new ecospace occupation did not contribute to biodiversification[2], and actually suggest that biotic factors may have played an important role at the regional scale in Baltica, but the fact that a major radiation takes place globally in the model without the need to invoke such ecological changes suggests that major global climate cooling may have played a leading role.

## Methods
### Climatic simulations

**Model description.** FOAM version 1.5[23] is a mixed-resolution ocean-atmosphere general circulation model. The atmosphere is a parallelized version of the National Center for Atmospheric Research's (NCAR) Community Climate Model 2 (CCM2), with radiative and hydrologic physics upgraded after CCM3 version 3.2[58]. It was run using a R15 spectral resolution (4.5° × 7.5°) and 18 levels in the vertical dimension. The oceanic component is the Ocean Model version 3 (OM3), a 24-level z-coordinate ocean general circulation model benefitting of a higher resolution than the atmosphere (1.4° × 2.8°). The simulation of sea ice uses the thermodynamic component of the CSM1.4 sea-ice model, which is based on the Semtner 3-layer thermodynamic snow/ice model[59]. The coupled model, FOAM, is well designed for paleoclimate studies. It has no flux correction and its quick turnaround time allows for long millennium-scale integrations. The higher execution speed compared to higher-resolution Earth system models, e.g. of the Coupled Model Intercomparison Project phase 6 generation, permits exploring the parameter space considered here (more than twenty 2000-year ocean-atmosphere simulations). FOAM has been widely used in paleoclimate studies[26,55,60], including in the Ordovician[25,61–64].

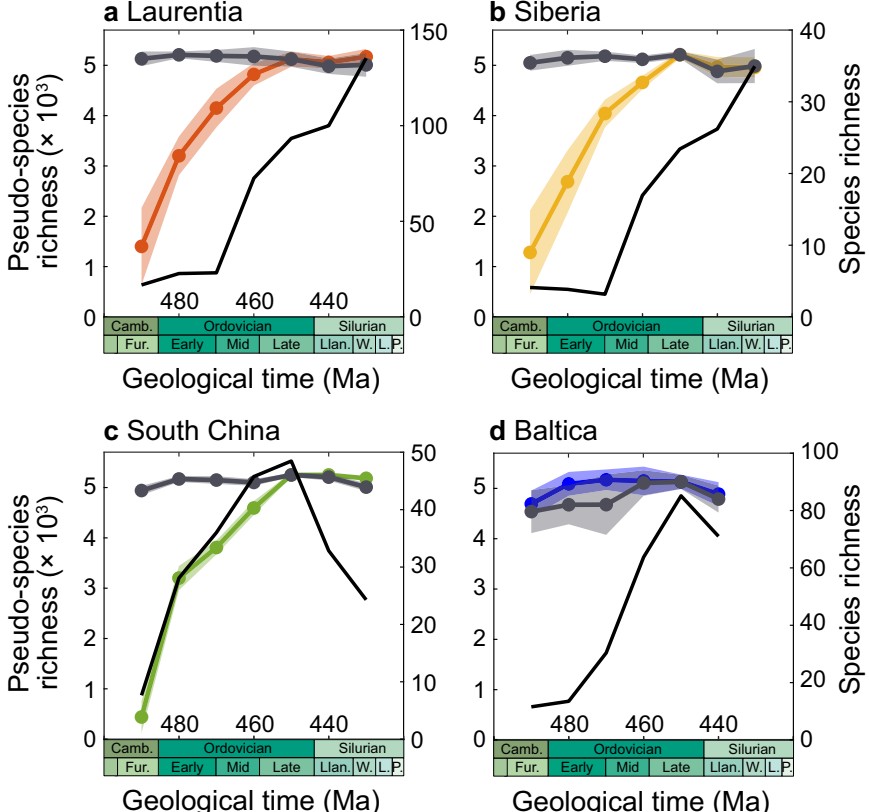

**Fig. 5 | Regional biodiversity.** Comparison of brachiopod fossil biodiversity[6] (black curves) with our simulations (curves with envelopes indicating median ±1 standard deviation) for Laurentia (**a**), Siberia (**b**), South China (**c**) and Baltica (**d**), for two scenarios as per Fig. 1: proxy-derived global climate cooling[10] (colored curves) and theoretical constant climatic state (sensitivity test to the continental configuration; gray).

**Description of the experiments.** We ran equilibrium FOAM simulations every 10 Myrs between 490 Ma and 430 Ma, both included (for a total of 7 time slices). In our main experiments, we used the Ordovician continental reconstructions of ref. 22. and a rocky desert vegetation type in every land grid point. Solar luminosity was calculated for each time slice after the stellar physics of ref. 24, increasing from 1313.24 W m$^{-2}$ at 490 Ma to 1319.7 W m$^{-2}$ at 430 Ma. In terms of astronomical forcing, we used a circular (null eccentricity) Earth's orbit and the obliquity was set to 22°. These parameters provide an equal, annual insolation for both hemispheres with minimal seasonal contrast. Simulations were run for a wide range of atmospheric $pCO_2$ values permitting to capture the magnitude of Ordovician cooling (Supplementary Fig. 1). In our main experiments, we reproduced the long-term Ordovician climatic evolution of ref. 10. by imposing atmospheric $pCO_2$ values of 192 PAL (preindustrial atmospheric $CO_2$ level; 1 PAL = 280 ppm $CO_2$) at 490 Ma, 96 PAL at 480 Ma, 48 PAL at 470 Ma, 24 PAL at 460 Ma, 8 PAL at 450 Ma, 6.1 PAL at 440 Ma and 4 PAL at 430 Ma. While $pCO_2$ values used for 460–430 Ma are well in line with values reconstructed for the early Paleozoic using long-term carbon cycle models[63,65,66], $pCO_2$ values used for older time slices may seem particularly elevated. Yet, we note that $CO_2$ is the only greenhouse gas varied in our model while variations in the concentration of other gases, and notably the reduced atmospheric oxygen concentration[67], may have also contributed to global climate warming. Therefore, the $pCO_2$ forcing used should be considered as a $pCO_2$-equivalent forcing rather than strictly $pCO_2$. In addition, the temperature increase resulting from $CO_2$ forcing (i.e., climatic sensitivity) is highly model-dependent. For instance, even most recent general circulation models of the Coupled Model Intercomparison Project phase 6 (CMIP6) generation lead to a large spread of values ranging from 1.8 to 5.6 K per $pCO_2$ doubling[68]. The climatic sensitivity of FOAM is ca. 3.5 K[26]; lower

$pCO_2$ values would permit simulating identical early Paleozoic temperatures using a model with a higher climatic sensitivity. Finally, the exceptionally high climatic sensitivity and/or $pCO_2$ increase required to simulate early Paleozoic climate is not unexpected. It was previously noted by ref. 37 and it is not our objective here to provide climatic mechanisms. Instead, we focus on the consequence of resulting ocean temperatures for marine habitability and the temporal trajectory of marine biodiversity in the deep geological past.

Simulations were initialized with a spatially uniform field of ocean salinity (35‰). In order to ensure intense convection and rapid deep-ocean equilibrium, simulations were initialized with a warm ocean. Simulations were run for 2000 years until deep-ocean thermal equilibrium (median ocean temperature drift of 0.0035 °C yr$^{-1}$ over the last 100 years at 3700 m depth). Mean annual results from the last 50 years were used to derive the temperature forcing fields used in the biodiversity model.

## Macroecological modeling
**Model description.** We modeled species richness in the Ordovician from 490 Ma to 430 Ma with a 10-Myr interval by means of a model based on the MacroEcological Theory on the Arrangement of Life (METAL)[35,69]; the version of the model used here was Species Niche and Climate Integration (SNCI) model[35]. The METAL theory is described in https://biodiversite.macroecologie.climat.cnrs.fr. We created pseudo-species (i.e. virtual species), each having a unique thermal niche with distinct degrees of thermophily and eurythermy. No hypothesis was made about species distribution. We allowed pseudo-species to settle in any given region of the global ocean so long as they could withstand the mean annual upper-ocean temperatures averaged of the first 100 meters of the water column. Thermal niches within the pool ranged from $t_{min} = -1.8$ °C, the temperature at which seawater freezes, to

$t_{max} = 44\,°C$, the maximum temperature under which a metazoan can reproduce[70]. The breadth of the thermal niche varied from $n = 1\,°C$ (stenothermic species) to $n = 45.8\,°C$ (universal eurythermic species). In total, 101,397 thermal niches were generated, as a function of niche breadth and the increment between niche breadth. To model species richness, we adapted the procedure described in Zacaï et al. (ref. [20]). A total of 10,000 pseudo-species were randomly selected (without replacement) among the 101,397 thermal niches available. The random extraction was repeated 100 times. We used a lower number of repetitions compared to previous work (i.e., 100 vs. 1000 in ref. [20]) to reduce computational cost but results were poorly impacted by the number of repetitions.

**Description of the experiments.** We ran the SNCI model on temperatures simulated with the ocean-atmosphere general circulation model FOAM reproducing, in our main experiments, the temperature reconstruction of Song et al. (ref. [10]) (Supplementary Fig. 1a). In order to best capture the environmental conditions characterizing Ordovician shallow-water shelves, we used the upper-ocean temperatures averaged over the first 100 m. We mapped the results from 490 Ma to 430 Ma by calculating an average biodiversity based on the results of all members of the ensemble of 100 model realizations (Fig. 1). We then derived a global biodiversity estimate by calculating the median of all biodiversity values for each map between 490 Ma and 430 Ma. The median was chosen to avoid any strong alteration of the central tendency by potential outliers. We calculated the standard deviation of simulated biodiversity in the spatial domain to represent the likely range of values found in each time slice and represented the result in the form of an envelope (Fig. 2).

## Statistics & reproducibility
No data was collected for this study.

## Reporting summary
Further information on research design is available in the Nature Portfolio Reporting Summary linked to this article.

## Data availability
All material required to repeat the experiments, conduct the analysis of the results and generate the figures is available online[71] and assigned a https://doi.org/10.5281/zenodo.8307366. Biodiversity data used for model-data comparison were extracted from previously published work: global curves in Fig. 2 are after ref. 5, ref. 41. and ref. 3. Regional curves in Fig. 5 are after ref. 6.

## Code availability
All material required to repeat the experiments, conduct the analysis of the results and generate the figures is available online[71] and assigned a https://doi.org/10.5281/zenodo.8307366. The Matlab code is also available on GitHub (https://github.com/DanElie/Cooling-Oceans-Triggered-GOBE/tree/v1.2); this is the recommended mode of access as it will contain any updates and clarifications. Yannick Copin's code used for the Taylor Diagram in Fig. 3 is available on GitHub (https://gist.github.com/ycopin/3342888).

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

## Acknowledgements
Calculations were partly performed using HPC resources from DNUM CCUB (Centre de Calcul de l'Université de Bourgogne). This is a contribution of UMR 6282 Biogéosciences Team 'SEDS'. This is a contribution to UNESCO project IGCP 735 "Rocks and the Rise of Ordovician Life" (Rocks n' ROL). The authors acknowledge the support of the French *Agence Nationale de la Recherche* (ANR) under reference ANR-22-CE01-0003 (project ECO-BOOST; to G.B., B.L., T.S., A.P.) and of the programme TelluS of the *Institut National des Sciences de l'Univers*, CNRS (project ROSETTA; to G.B., B.L., T.S., A.P.). We also thank the CPER programme IDEAL (G.B.). This work was also supported by the graduate school IFSEA that benefits from grant ANR-21-EXES-0011 from the French National Research Agency, under the Investments for the Future Programme.

## Author contributions
A.P. and G.B. designed the study. D.E.O., G.B. and A.P wrote the manuscript with input from B.L., C.M.M. and T.S. C.M.M. produced the continental reconstructions. A.P. conducted the FOAM experiments. D.E.O. and G.B. conducted the METAL experiments. D.E.O. led the analysis of the model results.

## Competing interests
The authors declare no competing interests.
