## [Peer Review File · Nature Communications]

Impact of global climate cooling on Ordovician marine biodiversityEditorial Note: Parts of this Peer Review File have been redacted as indicated to remove third-party material where no permission to publish could be obtained.

REVIEWER COMMENTS

Reviewer #1 (Remarks to the Author):

The paper “Cooling Oceans Did Trigger Ordovician Biodiversification” is a well written, concise report of the results of a modelling of Ordovician marine biodiversity based on a combination of climate models, explicit paleogeographical reconstructions, and spatially explicit sea water temperature-species richness models. The results of the modelling largely reproduce global temporal diversity curves and Ordovician latitudinal biodiversity gradients, which were previously and independently estimated from empirical data.

The authors interpret their results as support for the hypothesis that global cooling triggered the biodiversification during the Ordovician. The presented results support their interpretation.

This is an interesting, novel, and significant approach to test among the hypotheses about the mechanism behind the Ordovician Biodiversification. It should be published in Nature Communications, not only because it discusses the processes that led to this specific radiation in the marine realm, but because it is a contribution toward the wider discussions about the geological history and evolution of the latitudinal diversity gradient.

That said, I would suggest a few improvements and I found some minor flaws in the text which are listed below. I would suggest, to refer more explicitly to the wider discussions about the nature and evolution of the latitudinal diversity gradient (see listed references below). Also, I would suggest a more informed discussion about potential processes that led to discrepancies between the model results and empirical data.

p. 4 line 83: Here it reads: “This spatial-temporal pattern arises from the niche-environment interaction”. Can you explain what you specifically mean with this? What interacts with the environment. I guess it is not the niche, because the niche is an abstract concept, if anything interacts with the environment, then the organism. So, in your model: what interacts?

p. 5 lines 99-101. It is said that the simulations result in a monotone richness increase in response to climate cooling. In the context of your model, can you say a few words about the causes of the increase in your model? Do more nodes simply correspond to livable thermal niches?

p. 7. Lines 143-144: It is stated that “The distribution of fossil occurrences shows that this pattern has changed over long periods of time.” With a single reference that only gives an example from Ordovician microfossils. I think, there are more appropriate references, e.g.,: Badgley, C., T.M. Smiley, R. Terry, E.B. Davis, L.R.G. Desantis, D.L. Fox, S.S.B. Hopkins, T. Jezkova, M.D. Matocq, N. Matzke, J.L. McGuire, A. Mulch, B.R. Riddle, V.L. Roth, J.X. Samuels, C.A.E. Strömberg, and B.J. Yanites. 2017. Biodiversity and Topographic Complexity: Modern and Geohistorical Perspectives. *Trends in Ecology & Evolution*, 32: 211–226.

Crame, J.A. 2002. Evolution of taxonomic diversity gradients in the marine realm: a comparison of Late Jurassic and Recent bivalve faunas. *Paleobiology*, 28: 184–207.

Jablonski, D., S. Huang, K. Roy, and J.W. Valentine. 2017. Shaping the latitudinal diversity gradient: new perspectives from a synthesis of paleobiology and biogeography. *The American Naturalist*, 189: 1–12.

Mannion, P. D., Upchurch, P., Benson, R. B., & Goswami, A. (2014). The latitudinal biodiversity gradient through deep time. *Trends in ecology & evolution*, 29(1), 42-50.

Powell, M. 2009. The Latitudinal Diversity Gradient of Brachiopods over the Past 530 Million Years. *The Journal of Geology*, 117: 585–594.

Saupe, E.E., C.E. Myers, A. Townsend Peterson, J. Soberón, J. Singarayer, P. Valdes, and H. Qiao. 2019. Spatio-temporal climate change contributes to latitudinal diversity gradients. *Nature ecology & evolution*, 3: 1419–1429.

Song, H., S. Huang, E. Jia, X. Dai, P.B. Wignall, and A.M. Dunhill. 2020. Flat latitudinal diversity gradient caused by the Permian–Triassic mass extinction. *Proceedings of the National Academy of Sciences*, 117: 17578–17583.

It would be good to discuss the results presented in the manuscript with the main topics discussed in these reviews: Phanerozoic temperature change vs paleogeography. What is the contribution of the current manuscript to these discussions?

p. 8. Line 170. There is written that the timing of the biodiversity increase “matches the paleontological database 490 Ma and 450 Ma.” What does this mean? Do you mean

"matches that of results from estimations based on data from the PBDB". If yes: reference is needed. Also: What means the "Afterwards" in the same line. After what?

p. 9 line 184-187: Here I would refer to Penny, A. M., Hints, O., & Kröger, B. (2021).

Carbonate shelf development and early Paleozoic benthic diversity in Baltica: a hierarchical diversity partitioning approach using brachiopod data. *Paleobiology*, 1-21.

doi:10.1017/pab.2021.3.

Why: Penny et al. (2021) give a hypothesis why the diversity rose so extremely in Baltica:

"The results are consistent with carbonate shelf development as a driver of Ordovician regional diversification, both by enhancing broadscale differentiation between shallow- and deep-marine environments and by generating heterogeneous carbonate environments that allowed increasing numbers of brachiopod genera to coexist. However, temporal turnover also contributed significantly to apparent regional diversity, particularly in the Middle–Late Ordovician."

It would be interesting to discuss if this process also potentially contributed to the discrepancies you see between the model results and estimations from empirical data in Laurentia and Siberia. In this context see also:

Close, R. A., Benson, R. B., Saupe, E. E., Clapham, M. E., & Butler, R. J. (2020). The spatial structure of Phanerozoic marine animal diversity. *Science*, 368(6489), 420-424.

p. 9. Line 200: Reference needed.

p. 10. Lines 206-207. I am not sure if I understand that sentence. Does this suggest that the authors hypothesize that biological drivers were absent (but see above). Or that they are not necessary to explain the diversification in the light of the model (but see lines 173, 184). I would suggest to improve the final statement.

p. 17 line 377: is written: "The procedure was repeated 100 times, ten times less than the previous procedure". It is not clear what is the previous procedure. Please improve.

Reviewer #2 (Remarks to the Author):

In the submitted manuscript "Cooling Oceans Did Trigger Ordovician Biodiversification" Ontiveros et al. use simulations to reconstruct potential biodiversity patterns during the Great Ordovician Biodiversification Event (GOBE). A Fast Ocean Atmosphere Model was employed first to reconstruct spatially explicit patterns of paleotemperature, which then defined habitats that were allowed to be inhabited by virtual species. The results of the authors suggest that the decreasing temperature creates the modern-like latitudinal diversity gradient that did not exist previously - increasing overall diversity, which matches the GOBE observed in the fossil record.

These results are excellent in highlighting the patterns under the conditions and processes that the authors assumed, and they definitely need to be published and shared with the scientific community. Yet, these results imply specific assumptions (see below), that need to be stated explicitly and might indicate that the primary interpretation of the results (i.e. climatic changes caused the GOBE) needs to be toned down (e.g. climate alone could cause the GOBE). Although there are reasons suggesting that the assumptions behind the results hold - in my opinion - they do not conclusively falsify other potential drivers of the diversification event (e.g. evolutionary innovations). Nevertheless, the results of the study represent a significant advancement and (with some additional analyses and clarifications) will be important contributions to the study of the early Phanerozoic life and the GOBE.

The results of the study are based on two important aspects of the study: 1) the distribution of virtual species in the temperature spectrum and 2) the overall decrease of global temperature, which was tracked by the authors using the CO₂ concentration in the climate modelling.

1) Virtual species distributions

As the authors noted, the results are entirely dependent on the distribution of species (niches) in the temperature spectrum. Based on the description and the supplied code (Step5_niches.m), the method used to generate virtual species has an implicit assumption that all possible species niches are equally likely to exist. In this model, the expected number of species niches is distributed in the temperature spectrum similar to a Beta

distribution, with most species occurring around the mid domain of $\sim 23^{\circ}\text{C}$, with richness declining towards the colder and warmer end symmetrically. The result is that at temperatures close to the maximum limit the expected richness will be lower, which is projected to geographic space - thus creating the pattern of lower diversity in the tropical Cambrian.

In the modern oceans, however, the spatial diversity of taxa varies extensively over latitude and temperature (especially across taxa), and, as mentioned by Beaugrand et al (2018, cited in Zacaï et al 2021, on which the authors rely) a considerable number of niches can be vacant/unsaturated - decoupling diversity from the number of potential niches that are possible in an area. Species also do not occupy their entire thermal niches, making this connection even looser. The method therefore illustrates the number of potential niches, and I would state this explicitly for clarity and unambiguity (e.g. the cooling climate makes a higher number of niches available and therefore could cause the increase in diversity). Although showing the results based on the assumed richness \sim temperature relationship is very important, they are, nevertheless, based on hard assumptions. The authors also assumed that that this relationship did not change over time through evolution, but it is easy to imagine a scenario where more species adapt to the high temperatures (i.e. more colder niches remain unfilled) when such conditions are more widely distributed. It is also important to note that model was assessed for fishes, cetaceans, pinnipeds and planktonic organisms - and its applicability to well-fossilizable, benthic organisms that make up the majority of the fossil record might be different than for planktonic lifeforms (such as in Zacaï et al 2021).

The authors can consider assessing the sensitivity of the results to differences in the distribution of species in the temperature dimension, not only to the changes of the maximum limit, but to the shape and balance of the niche-distribution as well (e.g. maintaining the same t-max but making the distribution of the number of niches at a temperature more asymmetric, i.e. shift the mode). In addition to the through-time comparison of simulated and observed patterns of diversity, the analyses of the latitudinal diversity gradient in each time interval (as in Zacaï et al 2021) might help with the assessment of the model's adequacy and make its applicability to the scenarios more

credible. For the sake of clarity and more accurate interpretation by the reader, the authors can use "number of sampled niches" or something similar throughout the manuscript to better illustrate the difference between actual diversity and what the models imply. The virtual species (i.e. niches) are also not limited in dispersion (e.g in Hagen et al. 2021), which is necessary for the continent configuration to have a serious effect on diversity patterns via allopatric speciation. This aspect (i.e. dispersal limitation) can also be added to the model if the effect of continent configuration is to be ruled out as a causal factor in the diversification.

2) Temperature and climate modelling

Given the sensitivity tests run by the authors (stable climate, changing continental configuration), the long-term increase in diversity and the change of the LDG shape is entirely dependent on the temperature decrease. Since I am neither a climate modeller nor a geochemist, I cannot assess the robustness of the paleoclimate reconstructions themselves. The authors also tried an alternative cooling scenario (Extended Data Figure 2), in which the presence of the diversity increase seems to be dependent on a single time point (480Ma) and the pseudodiversity at 490Ma is comparable to those later in the Ordovician. This indicates that the sensitivity to this parameter might need to be tested more, maybe by considering additional reconstructions (see Grossman and Joachimski, 2022). In any case, this clearly illustrates the importance of the climate reconstruction in producing these results.

In the paragraph between lines 194 to 207 the authors argue that their results have implications on the climate, which I find somewhat circular: the presented results are entirely dependent on both whether there is and overall decrease in the global temperature and whether the assumptions about the number of niches ~ temperature relationship holds. In my opinion, if we did not know anything about the history of the climate, the increase in global diversity and the model's implications would yield only a very weak suggestion about a potentially cooling climate.

In conclusion, the results themselves are interesting, and these points do not interfere with

their soundness, merely with their interpretation. It is important to add to the discussion of early Phanerozoic life that decreasing temperature is capable of increasing global diversity when these assumptions hold, but the results are not conclusive evidence that cooling is the single or a more likely cause, and that the increase of global diversity by purely biotic processes can be ruled out.

The manuscript is written in clear and concise language. Besides these major points, I have only a couple of suggestions for the figures:

Fig. 1. Consider reversing the order of the panels (going from older to younger)

Fig. 2. Some color would be nice for this figure - and a legend would also be useful. Please add to the methods how the standard deviations of simulations was calculated. I assume that these come from the 100 replicates, but I am not sure (also the authors did not indicate whether the sampling from the virtual species pool was using replacement or not).

Fig. 3 The understanding of this figure is a bit difficult because it is not clearly stated between which variables was the correlation assessed. What is the observed/predicted variable? Is it the pseudodiversity values in every raster cell? Or the average for the latitude?

Fig. 5 Could also do better with a bit of color.

Notes on links and deposited material:

- I am very glad that the authors decided to deposit the code they used in the analysis.
- Even though the climate models do not form the backbone of the study, I would recommend the deposition all necessary information to rerun the climate models upon the eventual publication of these results.
- Consider the long-term deposition of this information on Line 366. My browser indicated that this link might be a security threat and there is no guarantee that this link will stay live for good (it is very annoying to click on link in a published paper just to find out that it is

dead.)

References:

Beaugrand, G., Luczak, C., Goberville, E., & Kirby, R. R. (2018). Marine biodiversity and the chessboard of life. *PLOS ONE*, 13(3), e0194006.

<https://doi.org/10.1371/journal.pone.0194006>

Grossman, E. L., & Joachimski, M. M. (2022). Ocean temperatures through the Phanerozoic reassessed. *Scientific Reports*, 12(1), Article 1. <https://doi.org/10.1038/s41598-022-11493-1>

Hagen, O., Flück, B., Fopp, F., Cabral, J. S., Hartig, F., Pontarp, M., Rangel, T. F., & Pellissier, L. (2021). gen3sis: A general engine for eco-evolutionary simulations of the processes that shape Earth's biodiversity. *PLoS Biology*, 19(7), e3001340.

<https://doi.org/10.1371/journal.pbio.3001340>

Zacai, A., Monnet, C., Pohl, A., Beaugrand, G., Mullins, G., Kroeck, D. M., & Servais, T. (2021). Truncated bimodal latitudinal diversity gradient in early Paleozoic phytoplankton. *Science Advances*, 7(15), eabd6709. <https://doi.org/10.1126/sciadv.abd6709>

Reviewer #3 (Remarks to the Author):

The manuscript by Ontiveros et al. presents a modelling study of the relationship between Ordovician global climate cooling and biodiversification.

As I cannot comment on the validity of the macroecological modelling, my comments focus mainly on the paleotemperature record.

For my evaluation, a more detailed description of the modelling parameters and modelling results is needed (see below). In particular, the modelled latitudinal temperature gradients which are essential for the modelled latitudinal biodiversity gradients should be documented and evaluated. In addition, some misunderstandings need be clarified.

Line 41: „ to modern-like values (ca. 30°C) in the Early and Middle Ordovician”. This statement is not correct, because oxygen isotope temperatures decrease during the Early Ordovician and most of the Middle Ordovician with temperatures during the Late Ordovician > 30°C, at least in case of the temperature reconstructions published by Grossman & Joachimski 2022 (Scientific Reports 12:8938). This recent compilation of oxygen isotope temperatures shows higher temperatures compared to Song et al. because the latter authors didn't apply a latitude correction for seawater $\delta^{18}O$.

Line 66-67: pCO_2 has been adjusted to model the Ordovician cooling trend. Unfortunately, the assumed change in pCO_2 is not documented. It would be interesting to assess how realistic the assumed atmospheric CO_2 levels or the Ordovician decrease in pCO_2 are.

Lines 80-81: it is argued that the maximum simulated biodiversity shifted from 60° S at 490 to 470 Ma to low latitudes at 450 Ma. What this means to me is that at 490 to 470 Ma, temperatures at 60°S and 40° N must have been considerably colder than in the hot tropics. What about the modelled latitudinal temperature gradient from the equator to 40° N or 60° S. Is this temperature gradient consistent with a relatively flat latitudinal temperature gradient of a greenhouse climate?

Line 86-88: Lewis & Eichenseer (2021) argue that a variable spatial coverage of data can have an impact on the reconstruction of GLOBAL temperature. However, the temperature records of Trotter et al. (2008) and Song et al. (2019) are low latitude temperature records (Song et al: 0 to 40° latitude). Low latitudes do not show large sea surface temperature differences, so variations in the spatial coverage of the data should not have a significant effect. In addition, the combined Ordovician carbonate (brachiopod) and apatite (conodont) $\delta^{18}O$ records (Grossman & Joachimski 2022), records restricted to 0 to 30° latitude, show a comparable temperature decrease. Thus, the argument that the low-latitude temperature cooling is underestimated does not seem justified. And citing Lewis and Eichenseer 2021 is not supportive.

Line 124 to 130: The experiments indicate that thermal limits > 65 °C suppress the impact of

global climate cooling on biodiversity. I have problems follow this, as upper thermal limits of 65 to 78 °C are completely unrealistic for metazoans (except certain archaea and bacteria). Upper thermal limits may be at 47° C, possibly reflecting an oxygen limitation mechanism of thermal tolerance and a thermal sensitivity of macromolecular structures (Pörtner 2002). So what do these experiments indicate? Or have I missed something?

Line 147 to 149: The authors state that in the late Cambrian to Early Ordovician, the maximum modelled biodiversity is at mid to high latitudes. This implies that the tropics were hot and the mid to high latitudes were colder. How steep is the modeled latitudinal temperature gradient? Is this gradient realistic for 0 to 40° N or 0 to 60° S in a greenhouse world with ice-free poles (see comment above)? Unfortunately no information is given. This is crucial because in case of a too steep or unrealistic latitudinal temperature gradient, the whole modeling approach becomes questionable. I recommend that the authors provide this information in the supplementary material.

Line 197 to 199: “an assumption supported by clumped isotope measurements that show no long-term trend in seawater $\delta^{18}\text{O}$ over the last 500 Myrs”. This sentence is misleading because clumped isotopes are independent of the oxygen isotope composition of seawater. Clumped isotopes only depend on temperature, e.g. during calcite precipitation.

Figure 4a and extended data Fig. 3: Why is the Katian missing on the x-axis?

Response to the reviewers [NCOMMS-22-52383-T]

Cooling Oceans Did Trigger Ordovician Biodiversification

– Line numbers in this response refer to the revised manuscript with no revision marks –

Reviewer #1 (Remarks to the Author):

The paper “Cooling Oceans Did Trigger Ordovician Biodiversification” is a well written, concise report of the results of a modelling of Ordovician marine biodiversity based on a combination of climate models, explicit paleogeographical reconstructions, and spatially explicit sea water temperature-species richness models. The results of the modelling largely reproduce global temporal diversity curves and Ordovician latitudinal biodiversity gradients, which were previously and independently estimated from empirical data. The authors interpret their results as support for the hypothesis that global cooling triggered the biodiversification during the Ordovician. The presented results support their interpretation.

This is an interesting, novel, and significant approach to test among the hypotheses about the mechanism behind the Ordovician Biodiversification. It should be published in Nature Communications, not only because it discusses the processes that led to this specific radiation in the marine realm, but because it is a contribution toward the wider discussions about the geological history and evolution of the latitudinal diversity gradient. That said, I would suggest a few improvements and I found some minor flaws in the text which are listed below. I would suggest, to refer more explicitly to the wider discussions about the nature and evolution of the latitudinal diversity gradient (see listed references below). Also, I would suggest a more informed discussion about potential processes that led to discrepancies between the model results and empirical data.

We thank Reviewer #1 for acknowledging the novelty and wide interest of our results, and for their comments. We provide a point-by-point response to all comments hereafter.

p. 4 line 83: Here it reads: “This spatial-temporal pattern arises from the niche-environment interaction”. Can you explain what you specifically mean with this? What interacts with the environment. I guess it is not the niche, because the niche is an abstract concept, if anything interacts with the environment, then the organism. So, in your model: what interacts?

We apologize for this shortcut. It is rather real eco-physiological constraints, captured by the concept of “niche”, that interact with the environment. For clarity, we reworded as follows: “*Biodiversity was simulated using a biodiversity model based on the interaction between many modelled pseudo-species and their environment*” (lines 69–70); also on lines 91–92 “*the interaction between the modelled pseudo-species and their environment*”.

p. 5 lines 99-101. It is said that the simulations result in a monotone richness increase in response to climate cooling. In the context of your model, can you say a few words about the causes of the increase in your model? Do more nodes simply correspond to livable thermal niches?

Yes. In the model, numerous niches are generated between minima and maxima temperature thresholds and a maximum number of niches overlap intermediate temperatures. This mechanism is referred to as ‘thermal mid-domain effect’ and was previously identified and described by Brayard et al. (2005). We added the following explanations on lines 94–98:

“This result, i.e., that global cooling triggers biodiversification in the model, arises in part from the niche-temperature interaction and a thermal mid-domain effect^{34,35}; ecological niches being defined between minimum and maximum temperature bounds, a maximum number of niches overlaps intermediate

temperatures, resulting in maximum biodiversity around 23 °C (ref.³⁵).”

We also clarified our statements:

- on lines 167–170: *“During the late Cambrian and early Ordovician (490 to 470 Ma), the maximum in modelled biodiversity is located over mid to high latitudes, contrasting with modern LBGs, while the lower latitudes are too warm to host diverse and abundant marine life (Fig. 4).”*
- on lines 170–172: *“In response to global climate cooling, the maximum in biodiversity gradually shifts to lower latitudes as large habitats gradually become livable for the modelled pseudo-species.”*

p. 7. Lines 143-144: It is stated that “The distribution of fossil occurrences shows that this pattern has changed over long periods of time.” With a single reference that only gives an example from Ordovician microfossils. I think, there are more appropriate references, e.g.,:

Badgley, C., T.M. Smiley, R. Terry, E.B. Davis, L.R.G. Desantis, D.L. Fox, S.S.B. Hopkins, T. Jezkova, M.D. Matocq, N. Matzke, J.L. Mcguire, A. Mulch, B.R. Riddle, V.L. Roth, J.X. Samuels, C.A.E. Strömberg, and B.J. Yanites. 2017. Biodiversity and Topographic Complexity: Modern and Geohistorical Perspectives. *Trends in Ecology & Evolution*, 32: 211–226.

Crame, J.A. 2002. Evolution of taxonomic diversity gradients in the marine realm: a comparison of Late Jurassic and Recent bivalve faunas. *Paleobiology*, 28: 184–207.

Jablonski, D., S. Huang, K. Roy, and J.W. Valentine. 2017. Shaping the latitudinal diversity gradient: new perspectives from a synthesis of paleobiology and biogeography. *The American Naturalist*, 189: 1–12.

Mannion, P. D., Upchurch, P., Benson, R. B., & Goswami, A. (2014). The latitudinal biodiversity gradient through deep time. *Trends in ecology & evolution*, 29(1), 42-50.

Powell, M. 2009. The Latitudinal Diversity Gradient of Brachiopods over the Past 530 Million Years. *The Journal of Geology*, 117: 585–594.

Saupe, E.E., C.E. Myers, A. Townsend Peterson, J. Soberón, J. Singarayer, P. Valdes, and H. Qiao. 2019. Spatio-temporal climate change contributes to latitudinal diversity gradients. *Nature ecology & evolution*, 3: 1419–1429.

Song, H., S. Huang, E. Jia, X. Dai, P.B. Wignall, and A.M. Dunhill. 2020. Flat latitudinal diversity gradient caused by the Permian–Triassic mass extinction. *Proceedings of the National Academy of Sciences*, 117: 17578–17583.

It would be good to discuss the results presented in the manuscript with the main topics discussed in these reviews: Phanerozoic temperature change vs paleogeography. What is the contribution of the current manuscript to these discussions?

We are grateful to Reviewer #1 for providing this list of references of interest. In the revised version of our manuscript, we support the statement that *“The distribution of fossil occurrences shows that this pattern has changed over long periods of time”* with additional references to the review by Mannion et al. (2014) and the Permian-Triassic case study by Song et al. (2020).

We also added a paragraph explaining the contribution of our results to the discussions on the latitudinal biodiversity gradient on lines 187–193:

“Our results, obtained with a coupled climate-macroecological model featuring minimal biological assumptions, shed new light on the mechanisms that allowed the establishment of a modern-like LBG for the first time in Earth’s history. They suggest that global climate change, rather than the two other usual suspects, paleogeography or history⁴⁸, was decisive. They also reaffirm the contribution of a cool global climatic state to the development of a steep LBG featuring a maximum in biodiversity at the low latitudes in line with paleontological data of the Permian-Triassic transition⁴⁷.”

p. 8. Line 170. There is written that the timing of the biodiversity increase “matches the paleontological database 490 Ma and 450 Ma.” What does this mean? Do you mean “matches that of results from estimations based on data from the PBDB”. If yes: reference is needed.

Yes, this is what we mean, thank you for inviting us to clarify. We reworded this way (lines 199–201):
“the general timing of biodiversity increase matches the temporal trends reconstructed based on data from the PBDB between 490 Ma and 450 Ma”.

We now further support this statement with a reference to Harper et al. (2021).

Also: What means the “Afterwards” in the same line. After what?

We meant *“From 440 Ma to 430 Ma”*. This is now clarified on line 201.

p. 9 line 184-187: Here I would refer to Penny, A. M., Hints, O., & Kröger, B. (2021). Carbonate shelf development and early Paleozoic benthic diversity in Baltica: a hierarchical diversity partitioning approach using brachiopod data. *Paleobiology*, 1-21. doi:10.1017/pab.2021.3.

Why: Penny et al. (2021) give a hypothesis why the diversity rose so extremely in Baltica: “The results are consistent with carbonate shelf development as a driver of Ordovician regional diversification, both by enhancing broadscale differentiation between shallow- and deep-marine environments and by generating heterogeneous carbonate environments that allowed increasing numbers of brachiopod genera to coexist. However, temporal turnover also contributed significantly to apparent regional diversity, particularly in the Middle–Late Ordovician.”

It would be interesting to discuss if this process also potentially contributed to the discrepancies you see between the model results and estimations from empirical data in Laurentia and Siberia. In this context see also:

Close, R. A., Benson, R. B., Saupe, E. E., Clapham, M. E., & Butler, R. J. (2020). The spatial structure of Phanerozoic marine animal diversity. *Science*, 368(6489), 420-424.

We thank Reviewer #1 for suggesting these interesting additions, which we included on lines 222–226:

“Model-data disagreement for Baltica may also reflect the inability of our model to capture finer, regional-scale environmental changes contributing to biodiversification. Indeed, quantitative analysis of brachiopod occurrences from the eastern Baltic paleobasin suggests that carbonate shelf development played a key role in regional diversification by enhancing substrate heterogeneity⁵², in line with the vision that reefs are biodiversity hotspots⁵³.”

p. 9. Line 200: Reference needed.

We added a reference to Hearing et al. (2018). Hearing et al. correct their Cambrian oxygen isotope measurements for the long-term Paleozoic trend (their Fig. 3). They suggest that such correction is necessary to obtain plausible temperature estimates in line with other greenhouse periods (their Fig. 4) and Cambrian climate models (their Fig. 5). We also refer to the seminal study of Veizer and Prokoph (2015).

p. 10. Lines 206-207. I am not sure if I understand that sentence. Does this suggest that the authors hypothesize that biological drivers were absent (but see above). Or that they are not necessary to explain the diversification in the light of the model (but see lines 173, 184). I would suggest to improve the final statement.

This is not our intention to claim that biological drivers were absent, but instead, as Reviewer #1 rightly states, that they are not necessary to explain the diversification in the light of the model. See line 122–124: *“Biological evolution may have played a role during the GOBE as well^{14,42}, but it is not necessary to invoke this process to simulate a first-order rise in marine biodiversity during the Ordovician in our model.”* We revised the final statement (lines 240–243):

“Our coupled climate-biodiversity modeling shows that a long-term cooling from very high temperature in the Cambrian to modern-like levels in the Ordovician provides a conservative, minimal scenario allowing the GOBE to take place in the model, and a modern-like LBG to emerge in agreement with paleontological evidence, even without invoking additional, biological drivers.”

p. 17 line 377: is written: The procedure was repeated 100 times, ten times less than the previous procedure". It is not clear what is the previous procedure. Please improve.

We improved the clarity of this explanation (lines 541–543):

"The random extraction was repeated 100 times. We used a lower number of repetitions compared to previous work (i.e., 100 vs. 1000 in ref.²⁰) to reduce computational cost but results were poorly impacted by the number of repetitions."

Reviewer #2 (Remarks to the Author):

In the submitted manuscript "Cooling Oceans Did Trigger Ordovician Biodiversification" Ontiveros et al. use simulations to reconstruct potential biodiversity patterns during the Great Ordovician Biodiversification Event (GOBE). A Fast Ocean Atmosphere Model was employed first to reconstruct spatially explicit patterns of paleotemperature, which then defined habitats that were allowed to be inhabited by virtual species. The results of the authors suggest that the decreasing temperature creates the modern-like latitudinal diversity gradient that did not exist previously - increasing overall diversity, which matches the GOBE observed in the fossil record.

These results are excellent in highlighting the patterns under the conditions and processes that the authors assumed, and they definitely need to be published and shared with the scientific community. Yet, these results imply specific assumptions (see below), that need to be stated explicitly and might indicate that the primary interpretation of the results (i.e. climatic changes caused the GOBE) needs to be toned down (e.g. climate alone could cause the GOBE). Although there are reasons suggesting that the assumptions behind the results hold - in my opinion - they do not conclusively falsify other potential drivers of the diversification event (e.g. evolutionary innovations). Nevertheless, the results of the study represent a significant advancement and (with some additional analyses and clarifications) will be important contributions to the study of the early Phanerozoic life and the GOBE.

We thank Reviewer #2 for their summary and comments, which are addressed below.

The results of the study are based on two important aspects of the study: 1) the distribution of virtual species in the temperature spectrum and 2) the overall decrease of global temperature, which was tracked by the authors using the CO₂ concentration in the climate modelling.

1) Virtual species distributions

As the authors noted, the results are entirely dependent on the distribution of species (niches) in the temperature spectrum. Based on the description and the supplied code (Step5_niches.m), the method used to generate virtual species has an implicit assumption that all possible species niches are equally likely to exist. In this model, the expected number of species niches is distributed in the temperature spectrum similar to a Beta distribution, with most species occurring around the mid domain of ~23°C, with richness declining towards the colder and warmer end symmetrically. The result is that at temperatures close to the maximum limit the expected richness will be lower, which is projected to geographic space - thus creating the pattern of lower diversity in the tropical Cambrian.

The model is based on the generation of many ecological niches between minimum and maximum temperature thresholds. It is true that a mid-domain effect arises in the Euclidean space of the niche. In our manuscript, we tested the sensitivity of our approach to the maximum temperature threshold (while the minimum thermal threshold is physically imposed by the temperature at which seawater freezes [-1.8°C]), thus altering the 'beta distribution' described by Reviewer #2. The resulting Taylor's diagram demonstrates that it is unlikely that the selection of another maximum threshold would substantially alter our conclusions (Fig. 3), confirming that the lower diversity in the tropical Cambrian ocean is a

robust model output that is poorly dependent on the exact shape of the beta distribution.

In the revised manuscript, we now clearly introduce the notion of ‘mid-domain’ effect on lines 94–98:

“This result, i.e., that global cooling triggers biodiversification in the model, arises in part from the niche-temperature interaction and a thermal mid-domain effect^{34,35}; ecological niches being defined between minimum and maximum temperature bounds, a maximum number of niches overlaps intermediate temperatures, resulting in maximum biodiversity around 23 °C (ref.³⁵).”

In the modern oceans, however, the spatial diversity of taxa varies extensively over latitude and temperature (especially across taxa), and, as mentioned by Beaugrand et al (2018, cited in Zacaï et al 2021, on which the authors rely) a considerable number of niches can be vacant/unsaturated - decoupling diversity from the number of potential niches that are possible in an area. Species also do not occupy their entire thermal niches, making this connection even looser. The method therefore illustrates the number of potential niches, and I would state this explicitly for clarity and unambiguity (e.g. the cooling climate makes a higher number of niches available and therefore could cause the increase in diversity).

Our model provides an assessment of the spatial distribution of species richness based on the generation of a large number of niches, with the assumption that a niche can only be occupied by a single species after the principle of exclusive competition of Gause (1964). Reviewer #2 is right in writing that there can be a decoupling between observed diversity and the number of potential niches available; the higher the biocomplexity of a group, the more likely the decoupling (Beaugrand et al., 2018). This decoupling originates from the fact that the spatial distribution of potential niches creates a mathematical constraints on the large-scale arrangement of biodiversity, which has been summarised by the allegory of the chessboard of life (Beaugrand, 2023; Beaugrand et al., 2018). Each taxonomic group has its own unique chessboard and niche saturation increases when organismal complexity decreases (Beaugrand, 2023). This is explained by basic ecological and evolutionary processes. Endosomatic energy decreases from primary producers to higher trophic levels as a consequence of the second law of thermodynamics, diminishing the number of individuals and therefore species richness and niche saturation from producers to top predators (Beaugrand, 2015; Lomolino et al., 2006). Having said that however, it has been shown that this chessboard of life strongly influences the contemporary arrangement of biodiversity from benthic to pelagic species and from protists to fish (Beaugrand et al., 2020). (Note that METAL also explains well the spatial arrangement of contemporary biodiversity from plants to mammals (Beaugrand et al., 2020; their Table 2)).

We added a summary of the above explanations on lines 73–79:

“In the model, thousands of pseudo-species were generated, each one occupying a unique thermal niche sensu Hutchinson²⁷ based on the principle of exclusive competition²⁸. Although the number of ecological niches can be decoupled from the number of species under certain circumstances in reality²⁹, we note that this and similar modeling approaches have previously been proven to satisfactorily simulate biodiversity at the global scale amongst groups of organisms of various complexity, both on land and in the ocean^{30,31}.”

In response to the Reviewer’s point, we now clearly state that the simulated increase in biodiversity actually corresponds to an increase in the number of niches available, on lines 88–89: *“the maximum of simulated biodiversity (or equivalently, maximum number of niches available) shifts...”* However, we think that the link between species richness and the number of niches is a technical point underlying our modelling and that referring to biodiversity changes this way would be more confusing than helpful to the reader. While revisions made in response to the Reviewer now explicitly define the relationship between niches and biodiversity on lines 73–79 and 88–89 (see above), we prefer using the term ‘biodiversity’ throughout the text for clarity.

Although showing the results based on the assumed richness ~ temperature relationship is very

important, they are, nevertheless, based on hard assumptions. The authors also assumed that that this relationship did not change over time through evolution, but it is easy to imagine a scenario where more species adapt to the high temperatures (i.e. more colder niches remain unfilled) when such conditions are more widely distributed. It is also important to note that model was assessed for fishes, cetaceans, pinnipeds and planktonic organisms - and its applicability to well-fossilizable, benthic organisms that make up the majority of the fossil record might be different than for planktonic lifeforms (such as in Zacaï et al 2021). The authors can consider assessing the sensitivity of the results to differences in the distribution of species in the temperature dimension, not only to the changes of the maximum limit, but to the shape and balance of the niche-distribution as well (e.g. maintaining the same t-max but making the distribution of the number of niches at a temperature more asymmetric, i.e. shift the mode).

In our approach, individual niches are generated between a minimum and a maximum temperature. We control the optimum and the range of the niches, as well as the degree of overlapping among niches. Once the niches are generated, this is the local thermal environment and its variability that determine the establishment of a niche and a pseudo-species. Therefore, the niche-environment interaction determines the resulting spatial patterns and the total number of pseudo-species. If we moved the niches towards the right (higher temperature) side of the thermal gradient, we would inflate the number of niches artificially and thereby increase niche packing at high temperatures. Such a simulation, which would generate a higher biodiversity at high temperatures due to an increase in the degree of niche overlapping, would be difficult to justify, however, because the increase in niche overlapping would also exacerbate the influence of the principle of competitive exclusion of Gause (1934). Therefore, we prefer not to include such additional simulations. Yet, we agree that the question of the biodiversity-temperature relationship is interesting. We now discuss this point on lines 98–104:

“In the absence of better constraints on the ecophysiology of marine organisms in the deep past, our approach necessarily relies on the assumption that this biodiversity–temperature relationship did not change over geological time. While such assumption may seem limiting, we note that the biodiversity–temperature relationship emerging from our model is supported by the mechanistic model of ref.³⁶ that demonstrates that the diversity-temperature relationship is expected, on grounds of thermodynamics and physiology principles, to peak at moderate temperatures in the range of 15–25°C and decrease outside of this range.”

In addition to the through-time comparison of simulated and observed patterns of diversity, the analyses of the latitudinal diversity gradient in each time interval (as in Zacaï et al 2021) might help with the assessment of the model's adequacy and make its applicability to the scenarios more credible.

We agree that it is important to ground truth our model with data. The comparison of our model results with the data of Zacaï et al. (2021) is provided on lines 178–181:

“simulated spatial-temporal patterns of biodiversity are supported by the LBG of Cambrian and Ordovician acritarchs, which show a minimum in biodiversity along the equator at 490 Ma and a shift of maximum biodiversity from ~60 °S at 480 Ma to the low latitudes during the Late Ordovician²⁰.”

Of course, we do not replicate the figure of Zacaï et al. (2021), but summarize the main trends. We also compare our simulated LBG with data of the PBDB based on the study of Kröger (2018) on lines 181–186:

“Our results are also supported by the LBG independently reconstructed during the Ordovician by Kröger et al. (ref.³²) based on the Paleobiology Database⁴, which shows a transition from a bimodal Early Ordovician LBG to a unimodal Late Ordovician LBG. The authors note that this evolution results from the large increase in low-latitude biodiversity during the GOBE, in agreement with the results of our coupled climate- macroecological model (Figs. 1, 4).”

For the sake of clarity and more accurate interpretation by the reader, the authors can use "number of sampled niches" or something similar throughout the manuscript to better illustrate the difference between actual diversity and what the models imply.

Although we now clearly explain that the number of pseudo-species corresponds to the number of niches available on lines 73–79 and 88–89, we prefer using the term ‘*biodiversity*’ throughout the text for clarity (see above) – but remain open to editorial decision.

The virtual species (i.e. niches) are also not limited in dispersion (e.g in Hagen et al. 2021), which is necessary for the continent configuration to have a serious effect on diversity patterns via allopatric speciation. This aspect (i.e. dispersal limitation) can also be added to the model if the effect of continent configuration is to be ruled out as a causal factor in the diversification.

This is an interesting point. However, allopatric speciation has a small impact on our results (Fig. R1). We explain this limited impact by the fact that the relatively short geological period (60 Ma) was not sufficient for main landmasses to move from fragmented to aggregated landmass configurations (Zaffos et al. 2017); at first order, and at our model resolution, the palaeogeographical configuration does not change much during the Ordovician (the response would probably be very different if we were interested in the whole Phanerozoic). The same general pattern remains during most of the studied period of time, with Gondwana occupying the South Pole and tropical landmasses at lower latitudes (Laurentia, Baltica and Siberia). The main difference observed between the simulations without and with allopatric speciation is the drop in global biodiversity at 430 Ma when accounting for allopatric speciation. This drop arises from the collision of Laurentia and Baltica, which together form Laurussia. It would be tempting to suggest that these new simulations capture to some extent the end-Ordovician extinction, but we prefer not to go that way because (1) the timing of the biodiversity drop is not very good and consequently (2) the biodiversity drop does not reflect the end-Ordovician mass extinction, which occurred before the collision between Laurentia and Baltica. Therefore, we do not think that these simulations represent an improvement and prefer to use the simple model without allopatric speciation. We think that the simplicity of our model constitutes a strength, and that increasing the model complexity with no clear added value would not be beneficial.

Figure R1: Total number of pseudo-species simulated without (blue) and with (red) allopatric speciation.

2) Temperature and climate modelling

Given the sensitivity tests run by the authors (stable climate, changing continental configuration), the long-term increase in diversity and the change of the LDG shape is entirely dependent on the temperature decrease. Since I am neither a climate modeller nor a geochemist, I cannot assess the robustness of the paleoclimate reconstructions themselves. The authors also tried an alternative cooling scenario (Extended Data Figure 2), in which the presence of the diversity increase seems to be dependent on a single time point (480Ma) and the pseudodiversity at 490Ma is comparable to those later in the Ordovician. This indicates that the sensitivity to this parameter might need to be tested more, maybe by considering additional reconstructions (see Grossman and Joachimski, 2022). In any case, this clearly illustrates the importance of the climate reconstruction in producing these results.

Reviewer #2 is completely right in emphasizing the importance of global climate change – this is our main goal here to show that global climate cooling is an important trigger of marine biodiversification during the Ordovician. We are grateful for suggesting the interesting addition of the cooling scenario of Grossman and Joachimski (2022). We conducted new simulations capturing this alternative, long-term cooling trend and added the new results in Supplementary Figs. 1 and 2. Because the temperatures reconstructed by Grossman and Joachimski (2022) are overall higher than in our 2 previous cooling scenarios, the resulting biodiversification taking place in the model is even stronger. This new scenario further strengthens our findings.

In the paragraph between lines 194 to 207 the authors argue that their results have implications on the climate, which I find somewhat circular: the presented results are entirely dependent on both whether there is and overall decrease in the global temperature and whether the assumptions about the number of niches ~ temperature relationship holds. In my opinion, if we did not know anything about the history of the climate, the increase in global diversity and the model's implications would yield only a very weak suggestion about a potentially cooling climate.

We understand the Reviewer's point. While we think that our results shed new light on the *consequences* of global climate cooling during the early Paleozoic, we acknowledge that our wording led to circularity. We revised the paragraph (lines 227–248). We renamed it "*Extrinsic vs. intrinsic drivers of GOBE*" and toned down the implications for early Paleozoic climate. We show key changes here:

- lines 228–230: "*Our results have implications for our understanding of the mechanisms that drove the GOBE. They highlight the decisive role of global climate change, although large uncertainties remain in the reconstruction of early Paleozoic ocean temperatures.*"
- lines 240–248: "*Our coupled climate-biodiversity modeling shows that a long-term cooling from very high temperature in the Cambrian to modern-like levels in the Ordovician provides a conservative, minimal scenario allowing the GOBE to take place in the model, and a modern-like LBG to emerge in agreement with paleontological evidence, even without invoking additional, biological drivers. We do not intend to mean that evolutionary innovation and new ecospace occupation did not contribute to biodiversification², and actually suggest that biotic factors may have played an important role at the regional scale in Baltica, but the fact that a major radiation takes place globally in the model without the need to invoke such ecological changes suggests that major global climate cooling may have played a leading role.*"

In conclusion, the results themselves are interesting, and these points do not interfere with their soundness, merely with their interpretation. It is important to add to the discussion of early Phanerozoic life that decreasing temperature is capable of increasing global diversity when these assumptions hold, but the results are not conclusive evidence that cooling is the single or a more likely cause, and that the increase of global diversity by purely biotic processes can be ruled out.

We revised our last paragraph and hope that these changes satisfactorily address the Reviewer's concerns.

The manuscript is written in clear and concise language. Besides these major points, I have only a couple of suggestions for the figures:

Fig. 1. Consider reversing the order of the panels (going from older to younger)

Done.

Fig. 2. Some color would be nice for this figure - and a legend would also be useful. Please add to the methods how the standard deviations of simulations was calculated. I assume that these come from the 100 replicates, but I am not sure (also the authors did not indicate whether the sampling from the virtual species pool was using replacement or not).

We changed the color of the main curve to blue, and added a chronostratigraphic chart with ages and colors after the International Chronostratigraphic Chart v2023/04. We also labelled the second y-axis (on the right).

We now clearly explain in the Methods how the standard deviation was calculated (lines 552–554): “We calculated the standard deviation of simulated biodiversity in the spatial domain to represent the likely range of values found in each time slice and represented the result in the form of an envelope”.

Finally, the sampling of the virtual species pool is without replacement. This is now stated on line 540.

Fig. 3 The understanding of this figure is a bit difficult because it is not clearly stated between which variables was the correlation assessed. What is the observed/predicted variable? Is it the pseudodiversity values in every raster cell? Or the average for the latitude?

We revised the caption of Fig. 3 to clearly state that (1) the modelled pseudo-species richness is assessed, and (2) it is considered in the 2D space (i.e., along the latitudes and longitudes). We further expanded the caption to explain more clearly how the figure is constructed and what it is used for.

“Figure 3. Sensitivity analysis to thermal maximum of pseudo-species by means of a Taylor diagram. We here compare the pseudo-species richness simulated in two dimensions (along the latitudes and longitudes) in various simulations. All simulations use the continental configuration for 460 Ma. The red star represents the standard simulation ran at 460 Ma with a tropical sea-surface temperature of ca. 30 °C (using $t_{max} = 44$ °C as the uppermost temperature for the pseudo-species thermal niches). This simulation is used as a reference. Numbers represent individual simulations ran (at 460 Ma as well) under a warmer climate (tropical sea-surface temperature of ca. 40 °C) featuring alternative t_{max} values ranging from 50 °C (simulation number 1) to 78 °C (simulation 10). The Taylor diagram, by combining representations of the correlation coefficient and the Root-Mean-Square Deviation (RMSD) between each simulation and the reference, and the standard deviation of pseudo-species richness in each simulation, allows us to investigate what values of t_{max} are required to make the pre-cooling and post-cooling pseudo-species richness similar at 460 Ma, i.e., what values of t_{max} are required to suppress the modelled increase in biodiversity in response to global climate cooling. It is used to demonstrate that the simulated increase in model biodiversity in response to cooling stands in the model, unless unlikely values are used for t_{max} (see main text).”

We also added a link to the code used for the Taylor diagram in the “Code availability” section, for transparency and reproducibility purposes.

Fig. 5 Could also do better with a bit of color.

We added some color to Fig. 5.

Notes on links and deposited material:

- I am very glad that the authors decided to deposit the code they used in the analysis.

- Even though the climate models do not form the backbone of the study, I would recommend the deposition all necessary information to rerun the climate models upon the eventual publication of these results.

Although we are engaged in open-access science – for instance the biodiversity model code being open-access along with every single model experiment being reproducible – we are also mindful that not every model code is fully open-access (e.g. the UK MetOffice climate model code that is widely used and published with is the property of the UK Ministry of Defense). The code of the FOAM model is the property of Rob Jacob (Argonne National Laboratory), nor ours, and we are consequently not allowed to make it publicly available. However, Rob Jacob does plan to upload his code on GitHub. In a recent email, he explained that he delayed things because he wants to preserve the change history which will require some thought about how to use svn2git on FOAM's old CVS repository. We are thus confident that the FOAM model code will be publicly available in the near future. In the meantime, and in an effort to make our work fully reproducible, we have uploaded all FOAM output data used in this study on Zenodo together with the biodiversity model (https://zenodo.org/record/8013420/preview/Cooling-Oceans-Triggered-GOBE-main.zip#tree_item11). (The link to the Zenodo repository is included in the Code Availability section.)

- Consider the long-term deposition of this information on Line 366. My browser indicated that this link might be a security threat and there is no guarantee that this link will stay live for good (it is very annoying to click on link in a published paper just to find out that it is dead.)

This is really just a link to the webpage of our co-author, Gregory Beaugrand. Although this webpage provides interesting, user-friendly additional information, all key data and code necessary to reproduce our study are freely accessible on Zenodo, and underlying theories and supporting explanations are provided in the scientific publications cited in this study. Put simply, this webpage provides interesting illustrations and explanations that may be of interest to our readers but, in the unlikely possibility that this webpage would not be supported in the long term, no critical information would be lost. We are not able to strictly guarantee the (very) long-term support of this url address; we think that keeping it here is good, but are open to deleting it to meet the journal guidelines.

References:

Beaugrand, G., Luczak, C., Goberville, E., & Kirby, R. R. (2018). Marine biodiversity and the chessboard of life. *PLOS ONE*, 13(3), e0194006. <https://doi.org/10.1371/journal.pone.0194006>

Grossman, E. L., & Joachimski, M. M. (2022). Ocean temperatures through the Phanerozoic reassessed. *Scientific Reports*, 12(1), Article 1. <https://doi.org/10.1038/s41598-022-11493-1>

Hagen, O., Flück, B., Fopp, F., Cabral, J. S., Hartig, F., Pontarp, M., Rangel, T. F., & Pellissier, L. (2021). gen3sis: A general engine for eco-evolutionary simulations of the processes that shape Earth's biodiversity. *PLoS Biology*, 19(7), e3001340. <https://doi.org/10.1371/journal.pbio.3001340>

Zacaï, A., Monnet, C., Pohl, A., Beaugrand, G., Mullins, G., Kroeck, D. M., & Servais, T. (2021). Truncated bimodal latitudinal diversity gradient in early Paleozoic phytoplankton. *Science Advances*, 7(15), eabd6709. <https://doi.org/10.1126/sciadv.abd6709>

Reviewer #3 (Remarks to the Author):

The manuscript by Ontiveros et al. presents a modelling study of the relationship between Ordovician global climate cooling and biodiversification. As I cannot comment on the validity of the macroecological modelling, my comments focus mainly on the paleotemperature record.

We thank Reviewer #3 for their comments on the paleotemperature record and note that Reviewer #2

provided independent evaluation of the modelling.

For my evaluation, a more detailed description of the modelling parameters and modelling results is needed (see below). In particular, the modelled latitudinal temperature gradients which are essential for the modelled latitudinal biodiversity gradients should be documented and evaluated. In addition, some misunderstandings need be clarified.

We provide a point-by-point response to all comments below.

Line 41: „ to modern-like values (ca. 30°C) in the Early and Middle Ordovician”. This statement is not correct, because oxygen isotope temperatures decrease during the Early Ordovician and most of the Middle Ordovician with temperatures during the Late Ordovician > 30°C, at least in case of the temperature reconstructions published by Grossman & Joachimski 2022 (Scientific Reports 12:8938). This recent compilation of oxygen isotope temperatures shows higher temperatures compared to Song et al. because the latter authors didn't apply a latitude correction for seawater $\delta^{18}O$.

We thank Reviewer #3 for referring to the work by Grossman and Joachimski (2022). It is an important contribution that provides an alternative cooling scenario for our simulations. In the revised version of the manuscript, we quantify the impact of this alternative scenario on our simulated biodiversity trend. Results, shown in Extended Data Figs. 1, 2, confirm that our results are robust when the temperature scenario is varied in the model.

While we now consider the temperature curve of Grossman and Joachimski (2022), we prefer keeping the reconstruction of Song et al. (2019) as our preferred scenario for our main experiments for the following reasons:

- Firstly, the reconstruction of Song et al. (2019) aligns better with the reconstruction of Goldberg et al. (2021) in terms of absolute values. The latter, recent reconstruction focuses on the early Paleozoic and is widely accepted in the early Paleozoic community. While Grossman and Joachimski (2022) correct their oxygen isotope data for ice volume based on first-order estimates of Phanerozoic glacial volumes extracted from a literature review, Goldberg et al. (2021) interestingly extract these important constraints from combined clumped- and oxygen isotope measurements. Additionally, while Grossman and Joachimski (2022) provide a first-order reconstruction of ocean temperatures at the Phanerozoic time scale, Goldberg et al. (2021) do specifically provide “a high-resolution record of early Paleozoic climate” – our time period of interest in the present contribution.

- Secondly, we note that many temperature curves are available for the Phanerozoic and that they largely differ in their absolute values due to the different databases and the various transfer function and assumptions selected when calculating paleotemperatures. While the curve of Song et al. (2019) falls well within the spread of the various curves, the reconstruction of Grossman and Joachimski (2022) constitutes an end member providing very high temperatures (see Fig. 5 of Grossman and Joachimski (2022)). A reason for the shift toward higher temperatures in Grossman and Joachimski (2022), compared with Song et al. (2019), is the choice of the phosphate-water paleothermometer of Pucéat et al. (2010), while Song et al. (2019) used the updated transfer function of Lécuyer et al. (2013; for a difference of up to -3.5 °C after Grossman and Joachimski (2022)).

Line 66-67: pCO_2 has been adjusted to model the Ordovician cooling trend. Unfortunately, the assumed change in pCO_2 is not documented. It would be interesting to assess how realistic the assumed atmospheric CO_2 levels or the Ordovician decrease in pCO_2 are.

We agree that this is an interesting point of discussion. We added the following explanations in the Methods (lines 499–518):

“In our main experiments, we reproduced the long-term Ordovician climatic evolution of ref.¹⁰ by imposing atmospheric pCO_2 values of 192 PAL (preindustrial atmospheric CO_2 level; 1 PAL = 280 ppm CO_2) at 490 Ma, 96 PAL at 480 Ma, 48 PAL at 470 Ma, 24 PAL at 460 Ma, 8 PAL at 450 Ma, 6.1 PAL at 440 Ma

and 4 PAL at 430 Ma. While $p\text{CO}_2$ values used for 460–430 Ma are well in line with values reconstructed for the early Paleozoic using long-term carbon cycle models^{63,65,66}, $p\text{CO}_2$ values used for older time slices may seem particularly elevated. Yet, we note that CO_2 is the only greenhouse gas varied in our model while variations in the concentration of other gases, and notably the reduced atmospheric oxygen concentration⁶⁷, may have also contributed to global climate warming. Therefore, the $p\text{CO}_2$ forcing used should be considered as a $p\text{CO}_2$ -equivalent forcing rather than strictly $p\text{CO}_2$. In addition, the temperature increase resulting from CO_2 forcing (i.e., climatic sensitivity) is highly model-dependent. For instance, even most recent general circulation models of the Coupled Model Intercomparison Project phase 6 (CMIP6) generation lead to a large spread of values ranging from 1.8 to 5.6 K per $p\text{CO}_2$ doubling⁶⁸. The climatic sensitivity of FOAM is ca. 3.5 K²⁶; lower $p\text{CO}_2$ values would permit simulating identical early Paleozoic temperatures using a model with a higher climatic sensitivity. Finally, the exceptionally high climatic sensitivity and/or $p\text{CO}_2$ increase required to simulate early Paleozoic climate is not unexpected. It was previously noted by ref.³⁷ and it is not our objective here to provide climatic mechanisms. Instead, we focus on the consequence of resulting ocean temperatures for marine habitability and the temporal trajectory of marine biodiversity in the deep geological past.”

Lines 80-81: it is argued that the maximum simulated biodiversity shifted from 60° S at 490 to 470 Ma to low latitudes at 450 Ma. What this means to me is that at 490 to 470 Ma, temperatures at 60°S and 40° N must have been considerably colder than in the hot tropics. What about the modelled latitudinal temperature gradient from the equator to 40° N or 60° S. Is this temperature gradient consistent with a relatively flat latitudinal temperature gradient of a greenhouse climate?

Reviewer #3 raises an interesting question here: is our model able to capture latitudinal temperature gradients? In previous work, we showed that “FOAM has been shown to satisfactorily capture the latitudinal seawater temperature gradient during both Paleozoic icehouse periods (Ordovician glaciation²⁵) and Mesozoic thermal maxima (Oceanic Anoxic Event 2, see Peer Review file published with ref.²⁶).” We included this additional information on lines 66–68. For illustration, we here reproduce the key figures of the references cited in Fig. R2.

[Figure redacted]

Figure R2: Case studies demonstrating that FOAM reasonably captures the seawater temperature latitudinal gradient in both icehouse and greenhouse conditions. Top panel (after Pohl et al., 2016): Comparison of the climatic belts reconstructed based on paleontological data during the Hirnantian glaciation, with the climatic belts simulated in FOAM at 3 PAL and 8 PAL – best match is obtained at 3 PAL. Bottom panel (after Wong Hearing et al., 2021 [Peer Review File published along with the article]): Comparison of the ocean temperature gradient simulated in FOAM at various pCO₂ levels and in the CMIP5-class model IPSL-CM5A2 at 4 PAL, with proxy data for Oceanic Anoxic Event 2. In FOAM, best model-data agreement is obtained at 4 PAL (or equivalently 1120 ppm atmospheric CO₂). Using this pCO₂ forcing, the latitudinal temperature gradient simulated in FOAM is very similar to the one simulated under similar boundary conditions using the recent CMIP5 model (compare panels C and D).

Line 86-88: Lewis & Eichenseer (2021) argue that a variable spatial coverage of data can have an impact on the reconstruction of GLOBAL temperature. However, the temperature records of Trotter et al. (2008) and Song et al. (2019) are low latitude temperature records (Song et al: 0 to 40° latitude). Low latitudes do not show large sea surface temperature differences, so variations in the spatial coverage of the data should not have a significant effect. In addition, the combined Ordovician carbonate (brachiopod) and apatite (conodont) δ18O records (Grossman & Joachimski 2022), records restricted to 0 to 30° latitude, show a comparable temperature decrease. Thus, the argument that the low-latitude temperature cooling is underestimated does not seem justified. And citing Lewis and Eichenseer 2021 is not supportive.

We respectfully disagree: the analysis of Lewis & Eichenseer (2021) confirms that Ordovician temperature proxy data come from low latitudes (their Fig. 3A), as noted by Reviewer #3. Yet, the latitudinal shift in available data does lead, if anything, and all other things kept unchanged, to a warming trend in reconstructed temperatures (their Fig. 4B,C). The authors note that “*the major Ordovician cooling trend observed from the δ18O temperature curve appears to be genuine [...]. In fact, the observed equatorward shift of sampling toward the end of the Ordovician suggests that cooling might be even more extreme than previously considered (e.g., Shields et al., 2003; Trotter et al., 2008)*”. Therefore, we think that the results of Lewis & Eichenseer (2021) do apply here. Although they do not impact our conclusion, they confirm that our assumptions of an important cooling trend during the Ordovician is robust and thus strengthen our study.

Line 124 to 130: The experiments indicate that thermal limits $> 65^{\circ}\text{C}$ suppress the impact of global climate cooling on biodiversity. I have problems follow this, as upper thermal limits of 65 to 78°C are completely unrealistic for metazoans (except certain archaea and bacteria). Upper thermal limits may be at 47°C , possibly reflecting an oxygen limitation mechanism of thermal tolerance and a thermal sensitivity of macromolecular structures (Pörtner 2002). So what do these experiments indicate? Or have I missed something?

We fully agree: an upper thermal limit of $> 65^{\circ}\text{C}$ is presumably unrealistic for metazoans. Actually, this is exactly the point we intend to make here: our conclusions (that cooling triggers biodiversification) are robust in the model, unless (presumably) unrealistic thermal limits are considered. We think that this analysis is crucial to demonstrate the robustness of our approach. We are making this point in the next paragraph on lines 152–161, where we state that “*Physiological adaptation of the Paleozoic Life to such extent is unlikely*”, provide arguments to support this statement, and finally conclude that “*the increase in biodiversity simulated in response to global cooling stands as a robust model result that is poorly dependent on (reasonable) changes in organism physiology.*” We think that this paragraph is relatively clear but are of course open to suggestion to improve it.

Line 147 to 149: The authors state that in the late Cambrian to Early Ordovician, the maximum modelled biodiversity is at mid to high latitudes. This implies that the tropics were hot and the mid to high latitudes were colder. How steep is the modeled latitudinal temperature gradient? Is this gradient realistic for 0 to 40°N or 0 to 60°S in a greenhouse world with ice-free poles (see comment above)? Unfortunately no information is given. This is crucial because in case of a too steep or unrealistic latitudinal temperature gradient, the whole modeling approach becomes questionable. I recommend that the authors provide this information in the supplementary material.

In response to a previous points made by Reviewer #3, we demonstrated that FOAM does satisfactorily capture latitudinal temperature gradients. Our previous studies conducted on the latest Ordovician glaciation and Cretaceous Oceanic Anoxic Event 2 support this statement (lines lines 66–68; see Fig. R2). We further note that another study conducted on Cambrian series 2 (ca. 510 Ma) documented temperature proxy data for Avalonia at ca. 65°S , and showed that FOAM simulates sea-surface temperatures compatible with such estimates using best-guess pCO_2 estimates for the period of time (Fig. R3; Hearing et al., 2018).

[Figure redacted]

Figure R3: Sea-surface temperatures simulated with FOAM at 510 Ma, compatible with the

temperatures reconstructed based on proxy data for Avalonia at ca. 65°S (black dot). After Hearing et al. (2018).

Line 197 to 199: “an assumption supported by clumped isotope measurements that show no long-term trend in seawater $\delta^{18}\text{O}$ over the last 500 Myrs”. This sentence is misleading because clumped isotopes are independent of the oxygen isotope composition of seawater. Clumped isotopes only depend on temperature, e.g. during calcite precipitation.

We apologize for the shortcut. Since clumped isotopes permit to reconstruct temperatures independently of the seawater composition, their coupling with oxygen isotopes permits to disentangle the contributions of temperature and seawater composition to $\delta^{18}\text{O}$ changes, and thus to calculate the oxygen composition of seawater. We reworded as follows: “an assumption supported by coupled oxygen isotope and clumped isotope measurements that show no long-term trend in seawater $\delta^{18}\text{O}$ over the last 500 Myrs⁵⁴” (lines 233–235).

Figure 4a and extended data Fig. 3: Why is the Katian missing on the x-axis?

We corrected the x-axis and took the opportunity to add colors in Figs. 2, 4 and 5 and Supplementary Figs. 2 and 3.

References cited

- Beaugrand, G., 2023. Towards an Understanding of Large-Scale Biodiversity Patterns on Land and in the Sea. *Biology* 12, 339. <https://doi.org/10.3390/biology12030339>
- Beaugrand, G., 2015. *Marine Biodiversity, Climatic Variability and Global Change*, 1st ed. Routledge.
- Beaugrand, G., Kirby, R., Goberville, E., 2020. The mathematical influence on global patterns of biodiversity. *Ecology and Evolution* 10, 6494–6511. <https://doi.org/10.1002/ece3.6385>
- Beaugrand, G., Luczak, C., Goberville, E., Kirby, R.R., 2018. Marine biodiversity and the chessboard of life. *PLoS ONE* 13, e0194006. <https://doi.org/10.1371/journal.pone.0194006>
- Brayard, A., Escarguel, G., Bucher, H., 2005. Latitudinal gradient of taxonomic richness: Combined outcome of temperature and geographic mid-domains effects? *Journal of Zoological Systematics and Evolutionary Research* 43, 178–188. <https://doi.org/10.1111/j.1439-0469.2005.00311.x>
- Gause, G.F., 1964. *The Struggle for Existence*.
- Harper, D.A.T., Cascales-Miñana, B., Kroeck, D.M., Servais, T., 2021. The palaeogeographical impact on the biodiversity of marine faunas during the Ordovician radiations. *Global and Planetary Change* 207, 103665. <https://doi.org/10.1016/j.gloplacha.2021.103665>
- Hearing, T.W., Harvey, T.H.P., Williams, M., Leng, M.J., Lamb, A.L., Wilby, P.R., Gabbott, S.E., Pohl, A., Donnadieu, Y., 2018. An early Cambrian greenhouse climate. *Science Advances* 4, eaar5690. <https://doi.org/10.1126/sciadv.aar5690>
- Kröger, B., 2018. Changes in the latitudinal diversity gradient during the Great Ordovician Biodiversification Event. *Geology* 46, 127–130. <https://doi.org/10.1130/G39587.1>
- Lomolino, M.V., Riddle, B.R., Brown, and J.H., 2006. *Biogeography, Third Edition, Third edition*. ed. Sinauer Associates, Inc.
- Mannion, P.D., Upchurch, P., Benson, R.B.J., Goswami, A., 2014. The latitudinal biodiversity gradient through deep time. *Trends in Ecology & Evolution* 29, 42–50. <https://doi.org/10.1016/j.tree.2013.09.012>
- Song, H., Huang, S., Jia, E., Dai, X., Wignall, P.B., Dunhill, A.M., 2020. Flat latitudinal diversity gradient

caused by the Permian–Triassic mass extinction. *Proceedings of the National Academy of Sciences* 117, 17578–17583. <https://doi.org/10.1073/pnas.1918953117>

Veizer, J., Prokoph, A., 2015. Temperatures and oxygen isotopic composition of Phanerozoic oceans. *Earth-Science Reviews* 146, 92–104. <https://doi.org/10.1016/j.earscirev.2015.03.008>

Zacai, A., Monnet, C., Pohl, A., Beaugrand, G., Mullins, G., Kroeck, D.M., Servais, T., 2021. Truncated bimodal latitudinal diversity gradient in early Paleozoic phytoplankton. *Science Advances* 7, eabd6709. <https://doi.org/10.1126/sciadv.abd6709>

REVIEWERS' COMMENTS

Reviewer #2 (Remarks to the Author):

Upon a second review, I still consider this paper very interesting and worthy of publication. Overall, I am happy with the authors' replies to my comments and glad to see them implemented.

I would still argue that there is a chance that the high number of species in the 15-25°C temperature range (same as the mid domain outlined by the min-maximum range) might still be the result of adaptation to the post-Cambrian climate and could be different in the past, potentially changing throughout the Cambrian-Ordovician interval. Again, this can happen without changing the maximum limit and by left-skewing the distribution (the 65°C upper limit, when the pattern breaks down, corresponds to a mode of ~33°C, which is warm, but perhaps not impossible). As the authors state in the rebuttal, this would mean a higher concentration of ranges in warmer temperatures. Competitive exclusion, as the authors state in their rebuttal, is only a problem if species have to occupy the same physical space, which (as the results of the authors demonstrate this) was larger in high-temperature areas. I simply disagree with the authors on the possibility of such a pattern, which is indeed less uniformitarian, but given that we are talking about totally different organisms, I think it remains an inexcludable eventuality.

There is no need to do any more analyses, but for the sake of completeness, I would welcome if the authors mentioned that this effect has the potential to influence the outcome of their study (the paragraph from Line 152 seems like a good place to do this).

I only have a couple of quick comments:

Line 62: "between 490 Ma 430 Ma, both included" - perhaps this can be simplified with "from 490 to 430 Ma".

Line 69: "which captures well biodiversity patterns" - consider reordering to "which captures biodiversity patterns well"

Figure 4. Note that the figure coloration itself is based on specific levels between contours, and the legend is based on continuous colors. The temporal continuity of the plot misleadingly suggests a time-continuous simulation, whereas the original results are 7 snapshots. Given that the quality of the patterns on the figure is not superb and the interpolated contours are very noisy, perhaps consider showing rectangular bands instead.

Figure 5. The lack of a visual legend makes this figure very difficult to read. Please also note that the envelopes that the authors mentioned are not present in the lower-resolution figure (they are present in the high-resolution ones at the end of the merged document). Also consider decreasing the point size.

I was also asked to provide feedback on the authors' response to Reviewer 1. I am glad that Reviewer 1 raised the awareness of discussion the latitudinal diversity gradient and I am content with the replies of the authors.

In any case, I would like to congratulate to the authors for the very exciting manuscript, which will be worthwhile contribution to the literature.

Response to the reviewers [NCOMMS-22-52383A]

Impact of global climate cooling on Ordovician marine biodiversity

– Line numbers in this response refer to the revised manuscript with no revision marks –

REVIEWERS' COMMENTS

Reviewer #2 (Remarks to the Author):

Upon a second review, I still consider this paper very interesting and worthy of publication. **Overall, I am happy with the authors' replies to my comments and glad to see them implemented.**

We thank Reviewer #2 for the time spent reviewing this new version of our manuscript.

I would still argue that there is a chance that the high number of species in the 15-25°C temperature range (same as the mid domain outlined by the min-maximum range) might still be the result of adaptation to the post-Cambrian climate and could be different in the past, potentially changing throughout the Cambrian-Ordovician interval. Again, this can happen without changing the maximum limit and by left-skewing the distribution (the 65°C upper limit, when the pattern breaks down, corresponds to a mode of ~33°C, which is warm, but perhaps not impossible). As the authors state in the rebuttal, this would mean a higher concentration of ranges in warmer temperatures. Competitive exclusion, as the authors state in their rebuttal, is only a problem if species have to occupy the same physical space, which (as the results of the authors demonstrate this) was larger in high-temperature areas. I simply disagree with the authors on the possibility of such a pattern, which is indeed less uniformitarian, but given that we are talking about totally different organisms, I think it remains an ineluctable eventuality. There is no need to do any more analyses, but for the sake of completeness, I would welcome if the authors mentioned that this effect has the potential to influence the outcome of their study (the paragraph from Line 152 seems like a good place to do this).

We included this discussion on lines 162–172:

“The temperature-biodiversity relationship also depends on the distribution of the niches in the temperature dimension. In our simulations, the niches are uniformly distributed along the temperature axis, between minimum and maximum temperatures. For a same t_{max} , however, an asymmetric distribution of the niches skewed towards the high temperatures would result in more species adapted to warmer climates. It could be argued that such adaptations of marine life to warmer temperatures may have developed in early Paleozoic oceans, significantly reducing the impact of global cooling on marine biodiversity in our simulations. However, such alteration of the temperature-biodiversity relationship would increase the number, packing and ultimately overlapping of niches at high temperatures. It would conflict with the fundamental principle of competitive exclusion²⁸. Therefore, we consider such changes in the temperature-biodiversity relationship as unlikely.”

I only have a couple of quick comments:

Line 62: "between 490 Ma 430 Ma, both included" - perhaps this can be simplified with "from 490 to 430 Ma".

Much better indeed. We corrected accordingly.

Line 69: "which captures well biodiversity patterns" - consider reordering to "which captures biodiversity patterns well"

Done.

Figure 4. Note that the figure coloration itself is based on specific levels between contours, and the legend is based on continuous colors. The temporal continuity of the plot misleadingly suggests a time-continuous simulation, whereas the original results are 7 snapshots. Given that the quality of the patterns on the figure is not superb and the interpolated contours are very noisy, perhaps consider showing rectangular bands instead.

We revised Fig. 4 by showing non-interpolated results ('rectangular bands') and also replaced the colormap with a colorblind-friendly gradient. We overlaid contours to facilitate the identification of the main patterns.

Figure 5. The lack of a visual legend makes this figure very difficult to read. Please also note that the envelopes that the authors mentioned are not present in the lower-resolution figure (they are present in the high-resolution ones at the end of the merged document). Also consider decreasing the point size.

We decreased the point size, but did not add any visual legend because space was limited (adding a legend would require an external panel) and we think that the caption gives the information necessary to understand the figure content. Regarding the envelopes, we apologize for the inconvenience and will make sure they appear in the proofs.

I was also asked to provide feedback on the authors' response to Reviewer 1. I am glad that Reviewer 1 raised the awareness of discussion the latitudinal diversity gradient and I am content with the replies of the authors.

We are happy to read that our responses satisfactorily addressed Reviewer #1's points.

In any case, I would like to congratulate to the authors for the **very exciting manuscript, which will be worthwhile contribution to the literature.**